# Axonal transport of Hrs is activity dependent and facilitates synaptic vesicle protein degradation

Veronica Birdsall[1],*, Konner Kirwan[1],*, Mei Zhu[2],*, Yuuta Imoto[3], Scott M Wilson[4] , Shigeki Watanabe[3,5], Clarissa L Waites[2,6]

**Turnover of synaptic vesicle (SV) proteins is vital for the maintenance of healthy and functional synapses. SV protein turnover is driven by neuronal activity in an endosomal sorting complex required for transport (ESCRT)-dependent manner. Here, we characterize a critical step in this process: axonal transport of ESCRT-0 component Hrs, necessary for sorting proteins into the ESCRT pathway and recruiting downstream ESCRT machinery to catalyze multivesicular body (MVB) formation. We find that neuronal activity stimulates the formation of presynaptic endosomes and MVBs, as well as the motility of Hrs+ vesicles in axons and their delivery to SV pools. Hrs+ vesicles co-transport ESCRT-0 component STAM1 and comprise a subset of Rab5+ vesicles, likely representing pro-degradative early endosomes. Furthermore, we identify kinesin motor protein KIF13A as essential for the activity-dependent transport of Hrs to SV pools and the degradation of SV membrane proteins. Together, these data demonstrate a novel activity- and KIF13A-dependent mechanism for mobilizing axonal transport of ESCRT machinery to facilitate the degradation of SV membrane proteins.**

## Introduction

Synaptic vesicles (SVs) are the fundamental units of neurotransmitter release, and their regulated fusion and recycling are essential for neuronal communication. These processes depend upon maintaining functional SV membrane proteins at the synapse. Indeed, deficits in SV protein turnover and degradation can precipitate synaptic dysfunction and neurodegeneration (Esposito et al, 2012; Bezprozvanny & Hiesinger, 2013; Hall et al, 2017). The complex morphology of neurons creates unique spatial challenges for SV protein clearance and degradation. For instance, SV membrane proteins are typically transported from presynaptic boutons to somatic lysosomes for degradation, while the machinery responsible for their degradative sorting is transported to boutons from cell bodies or more distal axons (Andres-Alonso et al, 2021; Roney et al, 2022). Neurons also face temporal challenges in transporting degradative machinery in response to stimuli such as synaptic activity. In dendrites, degradative organelles (proteasomes, lysosomes, and autophagosomes) undergo activity-dependent recruitment into spines as part of the mechanism for synaptic plasticity (Bingol & Schuman, 2006; Shehata et al, 2012; Goo et al, 2017) and must be rapidly mobilized to these sites. Neuronal activity also stimulates the turnover of SV and other presynaptic proteins (Sheehan et al, 2016; Truckenbrodt et al, 2018), requiring the local presence of degradative machinery to facilitate this process. However, very little is known about how neurons regulate the axonal transport and delivery of degradative machinery to presynaptic terminals.

Previous work from our group and others has demonstrated that the degradation of SV membrane proteins requires the endosomal sorting complex required for transport (ESCRT) pathway (Uytterhoeven et al, 2011; Sheehan et al, 2016). Comprising a series of protein complexes (ESCRT-0, -I, -II, -III, and Vps4), the ESCRT pathway recruits ubiquitinated membrane proteins and forms multivesicular bodies (MVBs) for delivery of this cargo to lysosomes (Hurley, 2015). Mutation or dysfunction of ESCRT and ESCRT-related proteins leads to neurodegeneration in animal models (Lee et al, 2007, 2019; Tamai et al, 2008) and is associated with frontotemporal dementia (FTD) (Skibinski et al, 2005; Lee et al, 2007), amyotrophic lateral sclerosis (Parkinson et al, 2006) and infantile-lethal epilepsy (Hall et al, 2017) in humans. Moreover, ESCRT machinery is essential for endosomal protein trafficking and implicated in the etiology of Alzheimer's and Parkinson's diseases (Spencer et al, 2014, 2016; Borland & Vilhardt, 2017; Vaz-Silva et al, 2018; Vagnozzi & Pratico, 2019; Feng et al, 2020). Despite the essential role of the ESCRT pathway in protein degradation and its links to neurodegenerative disease, little is known about its substrates, dynamic behavior, or mechanisms of localization and regulation in neurons.

[1]Neurobiology and Behavior PhD Program, Columbia University, New York, NY, USA  [2]Department of Pathology and Cell Biology, Columbia University Medical Center, New York, NY, USA  [3]Department of Cell Biology, Johns Hopkins University, Baltimore, MD, USA  [4]Department of Neurobiology, University of Alabama at Birmingham, Birmingham, AL, USA  [5]Solomon H Snyder Department of Neuroscience, Johns Hopkins University, Baltimore, MD, USA  [6]Department of Neuroscience, Columbia University, New York, NY, USA

Correspondence: cw2622@cumc.columbia.edu
*Veronica Birdsall, Konner Kirwan, and Mei Zhu contributed equally to this work.

Our recent work uncovered a pathway for activity-dependent degradation of SV membrane proteins, initiated by activation of the small GTPase Rab35 and its subsequent binding to ESCRT-0 protein Hrs, a Rab35 effector (Sheehan et al, 2016). Hrs is the initial component of the ESCRT pathway, responsible for sorting substrates into the pathway and recruiting downstream ESCRT machinery (Raiborg & Stenmark, 2009). As such, Hrs recruitment to presynaptic boutons likely functions as a rate-limiting step in SV protein degradation. In our previous study, we showed that elevating the firing rates of excitatory hippocampal neurons significantly increased the localization of Hrs to axons and SV pools (Sheehan et al, 2016), suggesting that Hrs synaptic recruitment is regulated by neuronal activity. In the current study, we use live imaging approaches to reveal the previously unexplored dynamics and mechanisms of Hrs axonal transport and synaptic delivery. We find that Hrs is transported on vesicles that exhibit activity-induced anterograde and bidirectional motility, co-transport ESCRT-0 protein (and Hrs binding partner) STAM1, and comprise a subset of Rab5+ early endosomes in axons. In addition, we identify kinesin motor protein KIF13A as essential for the activity-dependent transport of Hrs. Not only does neuronal activity stimulate the binding of KIF13A to Hrs, but KIF13A knockdown inhibits the activity-induced transport of Hrs along axons and its delivery to SV pools, as well as SV protein degradation. Together, these findings illuminate a critical mechanism by which ESCRT-0 proteins are transported to presynaptic boutons to facilitate SV protein degradation.

# Results

## Neuronal activity increases the formation of presynaptic endosomes

We previously showed that neuronal firing increases the localization of ESCRT-0 protein Hrs to axons and SV pools (Sheehan et al, 2016). Given the role of Hrs in recruiting downstream ESCRT machinery to catalyze MVB formation (Raiborg & Stenmark, 2009), we hypothesized that such Hrs recruitment would increase the number of MVBs and other endosomal structures at presynaptic boutons. We therefore examined electron micrographs of boutons from 16 to 21 day in vitro (DIV) hippocampal neurons treated for 48 h with the voltage-gated sodium channel blocker tetrodotoxin (TTX) to inhibit action potential firing, or the GABA receptor antagonist gabazine to increase firing (Fig 1A and B). Indeed, we observed a significantly higher density of presynaptic MVBs and endosomes (defined as single-membrane structures >80 nm in diameter; Fig 1B and C) following gabazine versus TTX treatment (Fig 1A, B, and E), suggesting that neuronal activity promotes the formation of these structures at boutons. Because other studies have reported that neuronal firing stimulates the biogenesis of autophagosomes at presynaptic terminals (Wang et al, 2015), we also counted the number of multilamellar structures present under each condition (Fig 1D and F). Interestingly, not only was the average density of these structures lower than that of endosomes/MVBs regardless of activity level (~0.1 multilamellar structures/$\mu m^2$ versus ~1 endosomes/$\mu m^2$ at baseline) but also their number did not increase with gabazine treatment (Fig 1F). These data indicate that prolonged

neuronal activity specifically increases the number of endosomes and MVBs at presynaptic boutons. To examine whether Hrs associates with such structures, we performed correlative-light electron microscopy in cultured hippocampal neurons coexpressing Halotag-Hrs and vesicular glutamate transporter (vGlut1-GFP) to label nearby SV pools. Indeed, we found Hrs labeling associated with endosomal structures including an MVB (Fig 1G, arrows), confirming its presence at putative sites of endolysosomal sorting.

## Neuronal activity stimulates the anterograde and bidirectional transport of Hrs

These ultrastructural findings, together with our previous work (Sheehan et al, 2016), suggest that the localization and transport of Hrs are regulated by neuronal activity. We thus investigated whether fluorescently- tagged Hrs exhibits activity-dependent changes in motility or directional transport in axons. To verify that overexpressed Hrs reflects the behavior of the endogenous protein, we immunostained 14–15 DIV hippocampal neurons with an Hrs antibody validated by immunoblot of tissue lysates from Hrs deficient mice (Fig S1A). Neurons were treated with either DMSO (control) or two pharmacological agents that increase action potential firing and were used in our previous study (Sheehan et al, 2016), the GABA receptor antagonist bicuculline and the voltage-activated potassium channel blocker 4-aminopyridine (Bic/4AP). We found that endogenous Hrs exhibited a punctate distribution along axons (Fig S1B), similar to the fluorescently- tagged protein (Sheehan et al, 2016). Moreover, 20 h of Bic/4AP treatment led to an increase in both the intensity and number of Hrs puncta per unit length of axon compared with the control condition (Fig S1C and D), consistent with our previous findings with RFP-Hrs.

We next performed imaging experiments in tripartite microfluidic chambers that isolate cell bodies from axons while enabling synapse formation between neurons in adjacent compartments (Taylor et al, 2005; Birdsall et al, 2019) (Fig S2A). Lentiviral transduction of cell bodies in these chambers led to moderate protein expression levels and consistent transduction efficiency across constructs used in this study (~50%; Fig S2B). Imaging of RFP-Hrs in 13–15 DIV neurons was performed either immediately following application of DMSO (control) or Bic/4AP (acute Bic/4AP), or 2 h after treatment with these drugs (2 h Bic/4AP) (Fig 2A–C; Videos 1 and 2), which did not cause any detectable changes in neuronal health over this timeframe (Fig S2C). Under control conditions, we observed that 38% of Hrs puncta were motile over the course of the ~4-min imaging session (Fig 2A, D, and F). Remarkably, acute treatment with Bic/4AP led to a slight but significant increase in puncta motility (to ~50% motile), with the effect growing stronger after 2 h of treatment (~60% motile puncta; Fig 2A–D and F). No change in speed was observed across the three conditions (Fig 2E), indicating that activity increases the number of Hrs puncta in the motile pool rather than changing the mechanism of transport. Surprisingly, while 2 h of TTX treatment decreased Hrs motility compared with Bic/4AP, this treatment did not decrease Hrs motility compared with the DMSO control condition (Fig S2D). These findings may reflect low levels of neuronal activity in the control condition, or variability in the motility of Hrs across the different batches of neuronal cultures that were combined for this analysis.

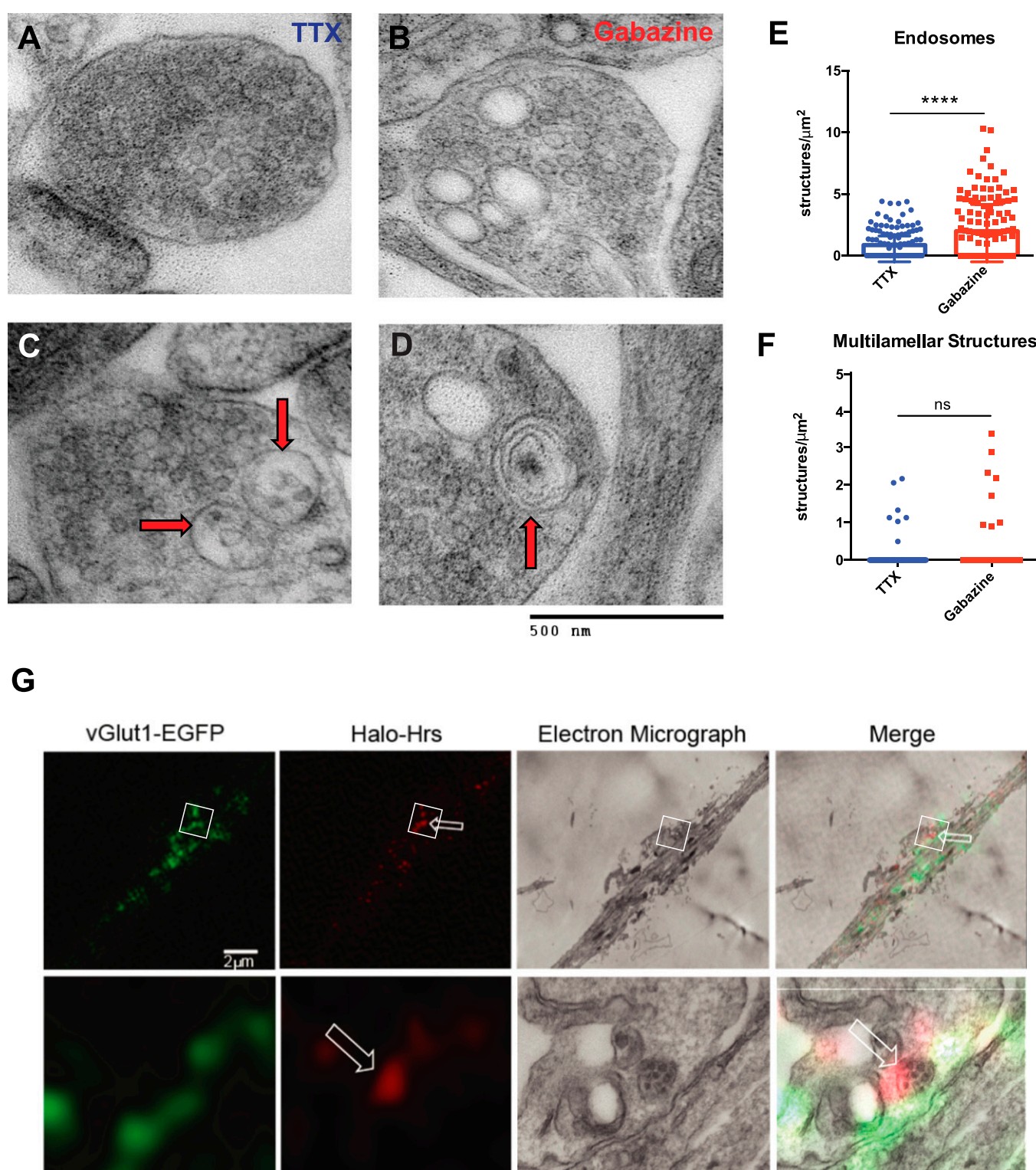

**Figure 1. Neuronal activity increases the number of presynaptic endosomes and Hrs associates with these structures.**
**(A, B)** Electron micrograph of presynaptic boutons from hippocampal neurons treated with TTX (A) or gabazine (B) for 48 h. **(C, D)** Electron micrographs of structures scored as endosomal (single-membrane, >80 nm diameter, red arrows; C) and multilamellar (>1 membrane, red arrows; D). Size bar, 500 nm. **(E, F)** Endosomal (E) and multilamellar (F) structures per μm² of presynaptic area at TTX- or gabazine-treated boutons. Scatter plot shows median and interquartile range (****$P < 0.0001$, ns $P = 0.8191$, Mann–Whitney test; n = 102 [TTX-treated] and 106 [gabazine-treated] boutons/condition). **(G)** Correlative-light electron microscopy images of hippocampal neurons expressing vGlut1-EGFP (green) and Halo-Hrs (Janelia-Halo-549 nm; red), treated for 2 h with Bic/4AP. White box (upper row) denotes region that is magnified in lower row; white arrow denotes Halo-Hrs–positive MVB. Note that images in lower row are rotated ~90° counterclockwise from corresponding upper images. Size bars are indicated on images.

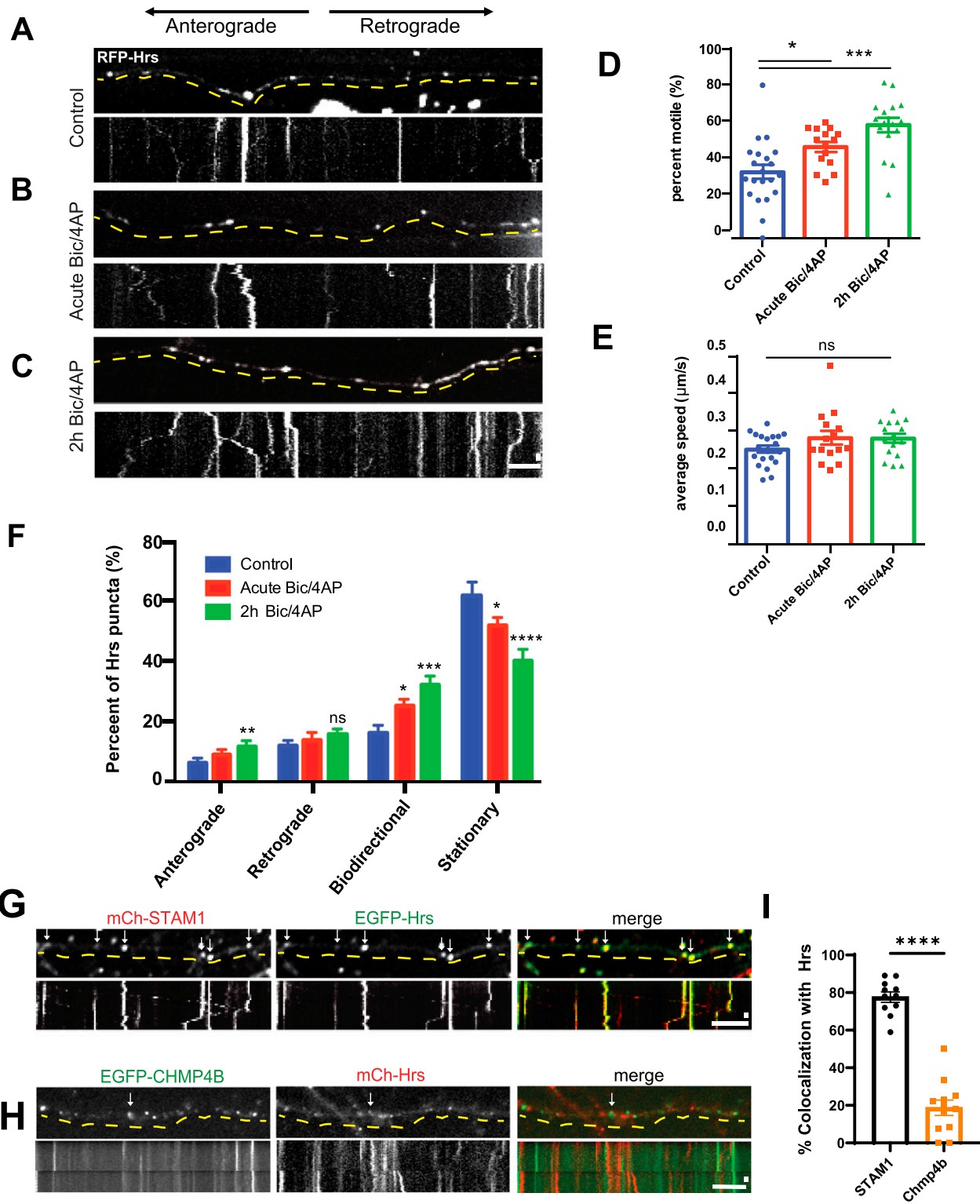

**Figure 2. Neuronal activity stimulates transport of Hrs+ vesicles in axons.**
**(A, B, C)** Images and corresponding kymographs of RFP-Hrs in microfluidically isolated axons from 13 to 15 day in vitro hippocampal neurons under control conditions (A; also see Video 1), acute treatment with Bicuculline/4AP (B), and after 2 h treatment with Bicuculline/4AP (C; Video 2). **(D)** Percentage of motile Hrs puncta in axons (****$P$ < 0.0001, *$P$ = 0.0198, one-way ANOVA with Dunnett's multiple comparisons test; ≥3 separate experiments, n = 20 [control], 15 [acute Bic/4AP], 16 [2 h Bic/4AP] videos/condition). **(D, E)** Average speed of motile Hrs puncta ($\mu$m/sec) (ns $P$ = 0.1594; same tests and "n" values as D). **(F)** Breakdown of directional movement of Hrs puncta, expressed as percentage of total Hrs puncta (****$P$ < 0.0001, ***$P$ = 0.0001, **$P$ = 0.0064, *$P$ = 0.0355 [bidirectional], *$P$ = 0.0198 [stationary], ns $P$ = 0.3102, one-way ANOVA with Dunnett's multiple comparisons test; same "n" values as D). **(G, H)** Images and corresponding kymographs for EGFP-Hrs and mCh-STAM1 (G) or EGFP-Chmp4b

Similar effects of Bic/4AP on axonal EGFP-Hrs motility were observed in dissociated hippocampal cultures (Fig S3A–E), demonstrating that these activity-induced motility changes are not artifacts of imaging in microfluidic chambers. Moreover, incubation of neurons with brain-derived neurotrophic factor (BDNF; 1 h), another stimulator of excitatory neuronal activity (Lessmann et al, 1994; Levine et al, 1995; Lessmann & Heumann, 1998), had the same effect as Bic/4AP on Hrs motility (Fig S3F–H), showing that this effect is not specific to Bic/4AP treatment.

Like other early endosome-associated proteins (e.g., Rab5) (Goto-Silva et al, 2019), Hrs exhibits a mixture of anterograde, retrograde, and non-processive bidirectional motility (Figs 2F–H and S3I). We examined whether neuronal activity influences the directional movement of Hrs, by measuring the net displacement of Hrs puncta over the 4-min imaging period (≥4 μm in a single direction for anterograde and retrograde motility, ≥4 μm total but <4 μm in a single direction for bidirectional, <4 μm total movement for stationary; also see the Materials and Methods section). Although bidirectional motility was the most strongly affected by Bic/4AP treatment, we also observed a small but significant increase in anterograde motility (Fig 2F). Further analysis revealed that the net displacement of Hrs puncta shifted slightly in the anterograde direction after 2 h of Bic/4AP (Fig S3J and K), suggesting that on average Hrs may be mobilized towards more distal axonal sites. Hrs is known to recruit STAM1, the other ESCRT-0 component, to form heterotetrameric ESCRT-0 complexes (Mayers et al, 2011; Takahashi et al, 2015). Therefore, we next investigated whether STAM1 is also present on Hrs+ vesicles. Coexpression of EGFP-Hrs and mCh-STAM1 in dissociated hippocampal neurons revealed that these proteins are cotransported in axons (~80% colocalization; Fig 2G and I and Video 3). In contrast, we observed very little cotransport of mCh-Hrs with EGFP-tagged ESCRT-III protein Chmp4b (~20% colocalization; Fig 2H and I). To ensure that Hrs overexpression did not influence STAM1's localization and dynamic behavior, we also evaluated mCh-STAM1 motility in singly transduced axons in microfluidic chambers. STAM1 dynamics were similar to those of Hrs, with increased motility after 2 h of Bic/4AP treatment, no change in average speed, and an increase in the fraction of anterograde and bidirectional puncta (Fig S4). Although we did not observe activity-dependent changes in Hrs retrograde transport (Fig 2F), we saw a significant increase in STAM1 retrograde movement after Bic/4AP treatment (Fig S4). This difference may reflect the higher expression and overall motility of STAM1 in axons, which made it easier to quantify changes in its directional movement. Together, these data show that the transport of both ESCRT-0 proteins is sensitive to neuronal firing.

### Hrs+ vesicles comprise a subset of Rab5+ vesicles in axons

ESCRT-0 proteins localize to early endosomes (Komada & Soriano, 1999; Raiborg et al, 2001; Sun et al, 2003; Raiborg & Stenmark, 2009; Flores-Rodriguez et al, 2015), and studies in non-neuronal cells show that Hrs recruitment to endosomal membranes depends upon interactions between its FYVE domain and the endosome-enriched lipid phosphoinositol-3-phosphate (PI3P) (Gaullier et al, 1998; Komada & Soriano, 1999; Stahelin et al, 2002). To determine whether this FYVE/PI3P interaction was similarly necessary for Hrs recruitment to axonal vesicles, we treated hippocampal neurons for 24 h with an inhibitor of PI3P synthesis, SAR405, and examined the number of EGFP-Hrs puncta per unit length axon as well as their fluorescence intensity. SAR405 treatment dramatically decreased both the density (~39%) and intensity (~36%) of EGFP-Hrs puncta (Fig S5A–C), indicating that PI3P is essential for Hrs association with axonal vesicles. This effect was replicated by an inactivating mutation of the Hrs FYVE domain (R183A; [Raiborg et al, 2001]) (Fig S5A–C), further confirming that the FYVE/PI3P interaction is critical for Hrs targeting to axonal vesicles. To evaluate whether the dynamics of the FYVE domain recapitulated those of full-length Hrs, we assessed the motility and directional movement of EGFP-2xFYVE in neurons cultured in microfluidic chambers under control, acute Bic/4AP, and 2 h Bic/4AP conditions. Interestingly, whereas the 2xFYVE domain exhibited punctate expression in axons, its overall motility was not affected by neuronal activity, although the proportion of retrogradely transported 2xFYVE puncta significantly decreased after acute Bic/4AP treatment (Fig S5D–G). These findings indicate that whereas the FYVE domain is required for Hrs' vesicle association, other domains are required for its activity-induced transport.

We next investigated whether activity-dependent motility is a general feature of early endosomes in axons. Here, we examined the motility of the small GTPase and early endosome marker Rab5a, critical for formation and function of the endolysosomal network (Gorvel et al, 1991; Deinhardt et al, 2006; Poteryaev et al, 2010), in axons from microfluidic chambers. We found that Rab5 puncta exhibited higher motility under control conditions than Hrs puncta (~60% motile puncta at baseline; Fig 3A and D and Video 4), with no increase in motility upon acute Bic/4AP application (Fig 3B and D) but a ~20% increase after 2 h of treatment (Fig 3C and D and Video 5). Similar to Hrs and STAM1, the average speed of Rab5 puncta did not change in response to neuronal firing (Fig 3E), but the proportion of anterogradely moving puncta increased significantly (Fig 3F). These data show that although axonal Rab5+ vesicles do exhibit activity-dependent changes in motility, this effect is more pronounced for ESCRT-0+ vesicles.

Because the axonal dynamics of Rab5 do not precisely align with those of ESCRT-0 proteins, we next investigated whether Hrs is present on the same vesicles as Rab5. Here, we performed time-lapse imaging of axons from dissociated neurons co-transfected with RFP-Hrs and EGFP-Rab5 (Fig 3G and Video 6). The density of Rab5 puncta in co-transfected axons was on average higher than that of Hrs puncta, and although >80% of motile Hrs puncta were also Rab5+ (Fig 3H), the converse was not true. Indeed, only ~60% of motile Rab5 puncta were Hrs+ (Fig 3I). These findings demonstrate that Hrs is present on a subset of Rab5+ structures, consistent with

and mCh-Hrs (H) in axons from dissociated hippocampal neurons under control conditions. Arrows show Hrs puncta that colocalize with STAM1 or Chmp4b.
**(I)** Colocalization of EGFP-Hrs with STAM1 and Chmp4b (****P < 0.0001, unpaired t test with Welch's correction, n = 11 [STAM1] and 13 [Chmp4b] videos/condition). For all images, horizontal size bar = 10 μm, vertical size bar = 25 s, 1 frame taken every 5 s (0.2 fps). Dashed yellow lines highlight axons. All scatter plots show mean ± SEM.

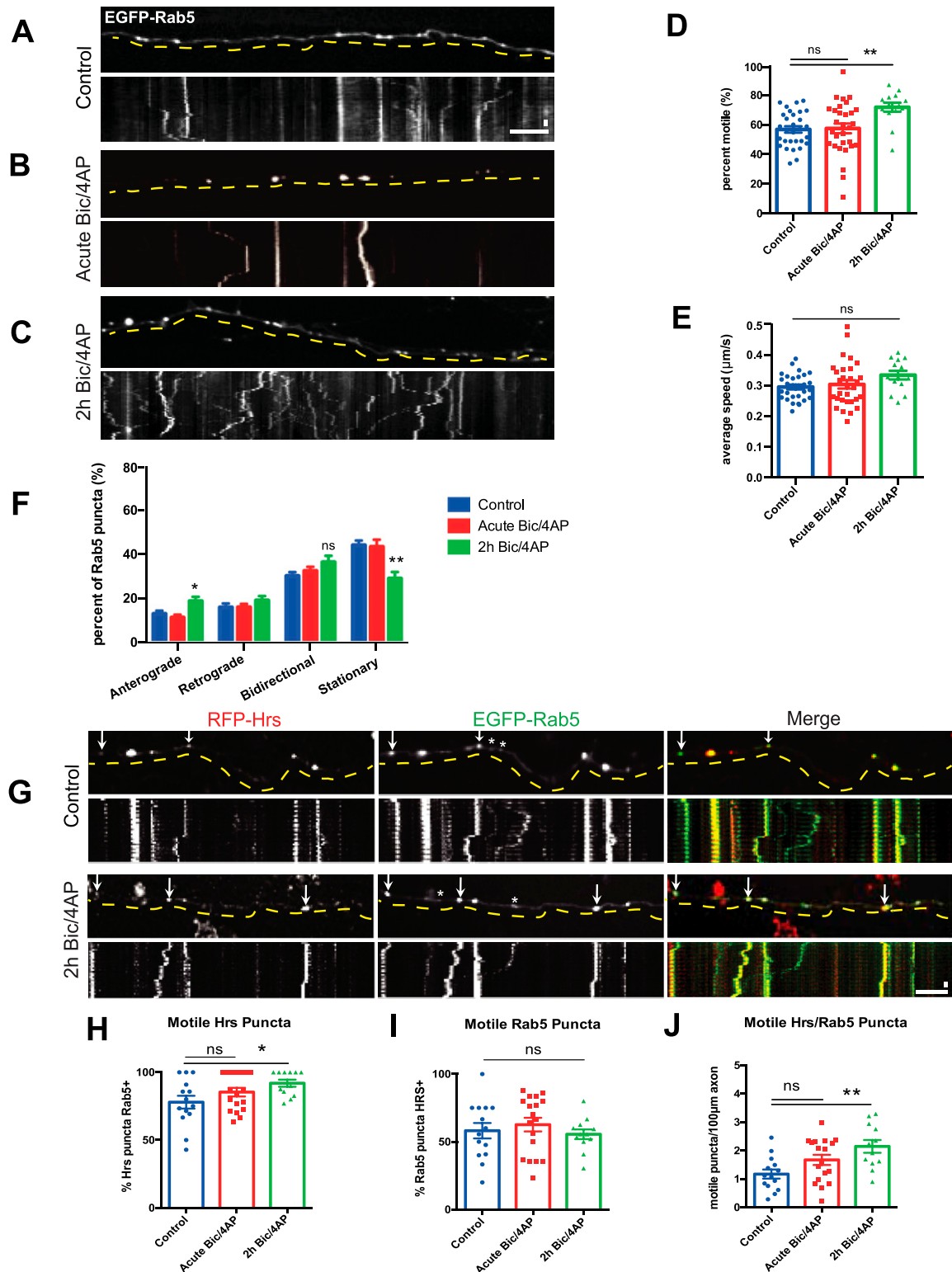

**Figure 3. Hrs is present on a subset of Rab5+ vesicles.**
**(A, B, C)** Images and corresponding kymographs of EGFP-Rab5 in microfluidically isolated axons from 13 to 15 day in vitro hippocampal neurons under control (A, see Video 3), acute Bic/4AP (B), and 2 h Bic/4AP (C, see Video 4) conditions. **(D)** Percentage of motile Rab5 puncta in axons (**$P$ = 0.0013, ns $P$ > 0.9999, Kruskal–Wallis test with Dunn's multiple comparisons; ≥3 separate experiments, n = 31 [control], 30 [acute Bic/4AP], 14 [2 h Bic/4AP] videos/condition). **(D, E)** Average speed of motile Rab5 puncta ($\mu m/sec$) (ns $P$ = 0.1295, same test and "n" values as D). **(F)** Breakdown of directional Rab5 puncta movement (**$P$ = 0.004, *$P$ = 0.0404, ns $P$ = 0.1082, one-way ANOVA with Dunnett's multiple comparisons test; same "n" values as D). **(G)** Images and corresponding kymographs of RFP-Hrs and EGFP-Rab5 in axons from dissociated hippocampal

previous studies showing that Rab5 associates with a heterogenous population of endosomes as well as SVs (Fischer von Mollard et al, 1994; Simonsen et al, 1998; Christoforidis et al, 1999; Shimizu et al, 2003; Flores-Rodriguez et al, 2015). However, the number of motile Hrs+/Rab5+ puncta increased significantly after 2 h Bic/4AP (Fig 3J), indicating that the Hrs+ subset of Rab5 vesicles is responsive to neuronal firing, even though the total population of Rab5 vesicles shows a more modest increase in activity-induced motility (Fig 3D).

Given that Rab35 mediates the recruitment of Hrs to SV pools and can colocalize with Hrs in axons (Sheehan et al, 2016), we also investigated whether Rab35 transport dynamics were similar to those of Hrs. As with STAM1 and Rab5, we performed time lapse imaging of EGFP-Rab35 in singly transduced axons in microfluidic chambers. Like Rab5, Rab35 puncta exhibited higher motility than Hrs at baseline; unlike Rab5 and ESCRT-0 proteins, these puncta did not exhibit any changes in motility or directional movement after acute or 2 h Bic/4AP treatment (Fig S6A–E). Moreover, the vast majority (~80%) of mCh-Rab35 puncta colocalize with the SV pool marker EGFP-Synapsin under control conditions, and this colocalization is unaffected by Bic/4AP treatment (Fig S6F and G), in contrast to Hrs ([Sheehan et al, 2016]; also see Fig 7). These findings indicate that the axonal localization and dynamics of Rab35 are distinct from those of Hrs.

### KIF13A interacts with Hrs in an activity-dependent manner

Our finding that Hrs puncta exhibit increased anterograde motility after 2 h Bic/4AP treatment (Fig 2F) implicates a plus-end directed kinesin motor in their activity-dependent transport. Previous studies identified several kinesins responsible for the anterograde transport of early endosomes, including kinesin-3 family members KIF13A and KIF13B (Huckaba et al, 2011; Bentley et al, 2015). Using an assay developed to evaluate the ability of these and other kinesins to transport cargo (Fig 4A) (Bentley & Banker, 2015), we examined whether KIF13A and/or KIF13B mediate Hrs transport. We first confirmed the efficacy of this assay in Neuro2a (N2a) cells via cotransfection of the FKBP-tagged dynein binding domain of BicD2 (BicD2-GFP-FKBP) and the FRB-tagged cargo binding domains of KIF13A or KIF13B (KIF13A-FRB-Myc, KIF13B-FRB-Myc). As previously reported, addition of membrane-permeant rapamycin analogue to link the FRB and FKBP domains led to minus-end-directed transport of BicD2 and KIF13A/B to the cell centrosome (Fig S7A and B). In addition, we were able to recapitulate transport of Rab5 by both KIF13A and KIF13B as previously reported (Fig S7C–F) (Bentley et al, 2015). We next tested the ability of KIF13A and KIF13B to facilitate transport of RFP-Hrs. Surprisingly, whereas Hrs was readily transported to the centrosome in the presence of linker and KIF13A, with 36% of transfected cells exhibiting this phenotype (Fig 4B, C, and F), it was rarely transported in the presence of linker and KIF13B

(~13% of cells; Fig 4D, E, and F). These findings indicate a specificity of the transport mechanism for Hrs+ vesicles that does not exist for Rab5+ vesicles. We further tested the specificity of interactions between Hrs and KIF13A/B by coimmunoprecipitation in N2a cells cotransfected with RFP-Hrs or mCherry control, together with the KIF13A- or KIF13B-FRB-Myc constructs. Consistent with our transport assays, we found that Hrs precipitates KIF13A to significantly greater extent than KIF13B (Fig 4G and H).

Because Hrs transport is regulated by neuronal activity, we next used the proximity ligation assay (PLA) to examine whether binding of Hrs to KIF13A is similarly regulated. After validation of the KIF13A antibody via immunoblotting and immunostaining of hippocampal neurons expressing shRNAs to knockdown KIF13A (shKIF13A1 and shKIF13A2; Fig S8A–D), we performed PLA to visualize interactions between EGFP-Hrs and endogenous KIF13A in the cell bodies of neurons under control or 2 h Bic/4AP conditions. In EGFP-expressing neurons, very few PLA puncta were present in the control condition, and neuronal activity did not change this value (Fig 5A and C), indicating little non-specific EGFP/KIF13A interaction under either condition. In contrast, EGFP-Hrs–expressing neurons exhibited a significant increase in PLA puncta after 2 h Bic/4AP treatment (Fig 5B and C), suggestive of an activity-dependent interaction between Hrs and KIF13A. To further confirm the activity dependence of this interaction, we performed co-immunoprecipitation experiments in N2a cells expressing RFP-Hrs and KIF13A-FRB-Myc after treatment with control or high K+ Tyrodes solution to mimic neuronal depolarization. Consistent with the PLA results, more KIF13A was co-immunoprecipitated with Hrs in cells treated with high K+ Tyrodes (Fig 5D and E). Together, these findings identify KIF13A as a candidate for the activity-dependent transport of Hrs.

### Knockdown of KIF13A inhibits activity-dependent Hrs motility

To test whether KIF13A is necessary for the axonal transport of Hrs, we examined the effect of KIF13A knockdown on Hrs motility in axons of 13–15 DIV neurons. For these studies, we used two shRNAs (shKIF13A1 and shKIF13A2) that led to ~50% knockdown of the protein when expressed from 2 to 15 DIV via lentivirus (Fig S8A and B). We first examined the effects of KIF13A knockdown on Hrs motility, by coexpressing EGFP-Hrs together with shKIF13A1 or a control shRNA (shCtrl) in neurons plated in microfluidic chambers. Time-lapse imaging revealed that expression of shKIF13A1 did not alter the motility (Fig 6A–C) or speed (data not shown) of Hrs puncta under control conditions compared with shCtrl. However, shKIF13A1 completely abolished the Bic/4AP-dependent increase in Hrs motility (Fig 6A–C and Videos 7–Videos 10). We further verified this effect with a second KIF13A hairpin, shKIF13A2 (Figs 6D and S8A, B, and G). In contrast, we saw no effect of KIF13B knockdown (by shKIF13B; Fig S8E–G) on the basal or Bic/4AP-dependent motility of

---

neurons under control and 2 h Bic/4AP conditions (see Videos 5 and 6). Arrows indicate motile puncta that are positive for Hrs and Rab5; stars indicate motile puncta that are only positive for Rab5. **(H)** Percentage of motile Hrs puncta that are Rab5+ (*P = 0.028, Kruskal-Wallis test with Dunn's multiple comparisons; ≥3 separate experiments, n = 14 [control], 18 [acute Bic/4AP], 12 [2 h Bic/4AP] videos/condition). **(H, I)** Percentage of motile Rab5 puncta that are Hrs+ (ns P = 0.5938, same test and "n" values as H). **(J)** Percentage of motile puncta that are Hrs+ and Rab5+ per 100 $\mu$m of axon (**P = 0.0025, ns P = 0.1023, one-way ANOVA with Dunnett's multiple comparisons test; same "n" values as H). For all images, horizontal size bar = 10 $\mu$m, vertical size bar = 25 s, 1 frame taken every 5 s (0.2 fps). Dashed yellow lines highlight axons. All scatter plots show mean ± SEM.

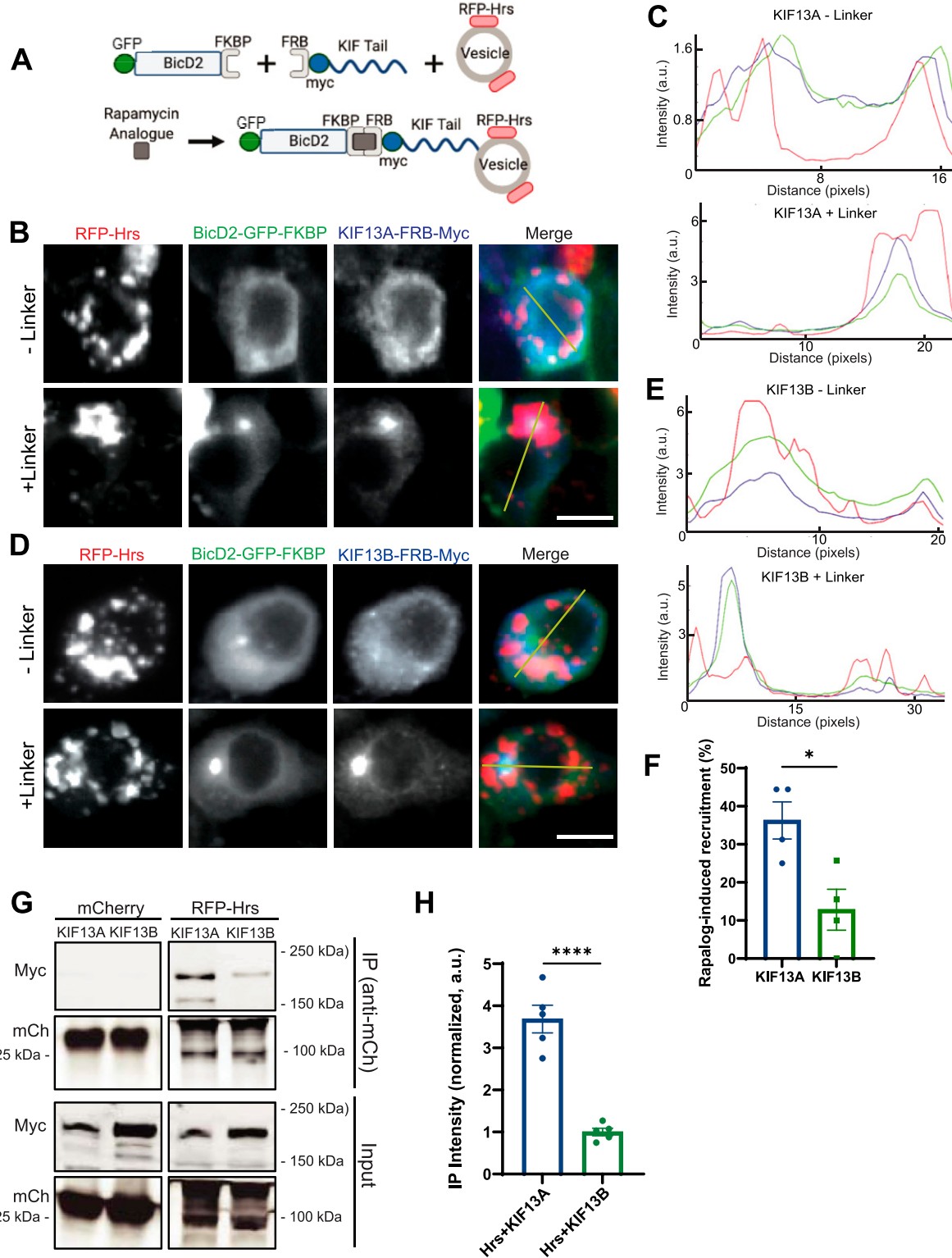

**Figure 4. Hrs is transported by KIF13A but not KIF13B.**
**(A)** Schematic diagram of transport assay using GFP/FKBP-tagged BicD2 motor domain, Myc/FRB-tagged kinesin cargo-binding tail domain, and fluorescently tagged cargo of interest. Upon addition of the rapamycin analogue (rapalog), FKBP and FRB interact and are co-transported to the centriole. If the protein/vesicle of interest interacts with the kinesin tail, it is also transported. **(B, C)** Images of N2a cells expressing RFP-Hrs (red), BicD2-GFP-FKBP (green), and KIF13A-FRB-Myc (blue), +/− rapamycin analogue to induce FKBP/FRB interaction (linker) (B), and corresponding line scan graphs (taken from left to right along the yellow lines in merged images; distance expressed in pixels) (C), showing that addition of linker induces colocalization of Hrs with BicD2 and KIF13A as depicted by a single peak in the lower graph. Size bar, 10 $\mu$m.

Hrs (Figs 6D and S8G). Finally, we found that KIF13A knockdown similarly prevented BDNF-induced motility of Hrs (Fig S8H and I), indicating that KIF13A is responsible for the activity-dependent transport of Hrs+ vesicles in axons.

We next investigated whether KIF13A is required for the activity-dependent delivery of Hrs to SV pools. Here, we monitored the colocalization of mCh-Hrs with EGFP-Synapsin in neurons transfected with shCtrl, shKIF13A1, or shKIF13A2 on five DIV and treated with vehicle control or Bic/4AP for 20 h on 13 DIV. In shCtrl neurons, Bic/4AP treatment significantly increased the colocalization of Hrs with Synapsin (Fig 7A and D), as anticipated based upon our previous findings (Sheehan et al, 2016). However, no such increase in colocalization was observed in neurons expressing shKIF13A1 or shKIF13A2 (Fig 7B–D), demonstrating that KIF13A is essential for the activity-induced recruitment of Hrs to SV pools. To determine whether this disruption of Hrs transport has an effect on SV protein degradation, we used our previously described cycloheximide-chase assay (Sheehan et al, 2016) to measure the degradation of SV membrane proteins SV2 and VAMP2 in 14 DIV neurons expressing shKIF13A1 versus shCtrl. Indeed, the degradation of both SV proteins was slowed in the presence of shKIF13A1 (Fig 7E–G). We further verified this finding via SNAPtag pulse-chase analysis (Bodor et al, 2012) in neurons expressing shCtrl or shKIF13A1 together with SNAP- and Flag-tagged VAMP2 (VAMP2-SNAP; used in our previous study [Sheehan et al, 2016]). Here, 13 DIV neurons were pulsed with Janelia Fluor 549 (JF549) for 30 min to label the existing pool of VAMP2-SNAP. After several washes, neurons were either fixed immediately (0 h timepoint) or 48 h post-labeling (48 h time point), and the percentage of total VAMP2-SNAP (detected by immunostaining with Flag antibody) labeled with JF549 was measured. In neurons expressing shCtrl, this percentage decreased significantly from the 0 to 48 h timepoint (~80% at 0 h versus ~50% at 48 h), indicating substantial degradation of pulsed VAMP2-SNAP (Fig 7H and I). In contrast, there was no significant change in this value for neurons expressing shKIF13A1 (Fig 7H and I), indicative of attenuated VAMP2-SNAP degradation. Together, these findings indicate that KIF13A is necessary for facilitating the degradation of SV membrane proteins, by delivering Hrs to presynaptic terminals.

# Discussion

In this study, we characterize the axonal transport of ESCRT-0 protein Hrs, uncovering a novel mechanism for its delivery to presynaptic boutons in response to neuronal firing (Fig 8). These findings provide the first example of activity-dependent regulation of degradative machinery in axons. Previous studies have reported that the axonal motility of mitochondria and the inter-bouton transport of SVs in hippocampal neurons are regulated by

neuronal activity, specifically presynaptic calcium influx (Chen & Sheng, 2013; Lin & Sheng, 2015; Gramlich & Klyachko, 2017; Qu et al, 2019). These processes are hypothesized to contribute to activity-dependent remodeling of synapses and to alter neurotransmitter release properties (Lin & Sheng, 2015; Gramlich & Klyachko, 2017). Synaptic activity has also been shown to induce the postsynaptic recruitment of proteasomes and lysosomes in cultured neurons, as well as the local biogenesis of autophagosomes at pre- and postsynaptic sites (Bingol & Schuman, 2006; Shehata et al, 2012; Wang et al, 2015; Soukup et al, 2016; Goo et al, 2017; Kulkarni & Maday, 2018). However, the axonal transport of autophagosomes in cultured neurons appears to be a largely constitutive process that is unaffected by stimuli such as synaptic activity and glucose levels (Maday et al, 2012; Maday & Holzbaur, 2012, 2016). Axonal autophagosomes, once formed, undergo reliable retrograde trafficking to the soma with their cargo while acidifying into autolysosomes (Maday et al, 2012; Maday & Holzbaur, 2012, 2016). In contrast, we find that axonal transport of ESCRT-0 proteins Hrs and STAM1 is bidirectional and highly sensitive to neuronal activity levels, likely facilitating delivery of this early ESCRT machinery to presynaptic sites to catalyze use-dependent protein degradation.

Our data suggest that Hrs+ vesicles represent a specialized endosomal subtype. Endosomes are highly dynamic and heterogenous structures in neurons (Maday & Holzbaur, 2016; Yap et al, 2018), reflecting the complex morphologies and protein trafficking demands of these cells. Early endosomes are classically defined by the markers Rab5 and EEA1 (Gorvel et al, 1991; Simonsen et al, 1998; Langemeyer et al, 2018); however, recent studies have identified sub-categories of early endosomes in neurons that can be differentiated both by their markers and axonal transport patterns (Leonard et al, 2008; Olenick et al, 2019). The Hrs+ vesicles that we describe here represent a novel addition to this category. We find that whereas the total pool of axonal Rab5+ vesicles exhibits high motility under baseline conditions and activity-induced increases in anterograde but not bidirectional motility, the Hrs+ subset of this pool exhibits limited motility at baseline, but activity-induced increases in both anterograde and bidirectional motility (Figs 2 and 3). Moreover, given the high degree of colocalization between Hrs and STAM1 (Fig 2), and the activity dependence of STAM1 retrograde transport (Fig S4), we think it possible that Hrs retrograde motility is similarly increased by activity, but that this increase is difficult to detect because of the less efficient expression of Hrs in neurons. Such an enhancement of anterograde and retrograde axonal transport during neuronal firing would increase the likelihood of ESCRT-0 delivery to/contact with SV pools to initiate activity-dependent SV protein degradation. This hypothesis, which will be further explored in future studies, aligns with previous research indicating that SV proteins become increasingly dysfunctional as

---

**(D, E)** Images of N2a cells expressing RFP-Hrs (red), BicD2-GFP-FKBP (green), and KIF13B-FRB-Myc (blue), +/– linker (D), and corresponding line scan graphs (taken along yellow lines in merged images) (E) showing that addition of linker does not induce Hrs colocalization with BicD2/KIF13B. Size bar, 10 μm. **(F)** Percentage of transfected cells exhibiting rapalog-induced Hrs recruitment to the centriole (*$P$ = 0.0178, unpaired $t$ test, n = 4 fields of view/condition). **(G)** Immunoblot of lysates from N2a cells coexpressing mCherry or RFP-Hrs together with KIF13A-FRB-Myc or KIF13B-FRB-Myc, immunoprecipitated (IP) with mCherry antibody and probed with Myc or mCherry antibodies. **(H)** Quantitative analysis of co-immunoprecipitated KIF13A/B, normalized to immunoprecipitated RFP-Hrs, and expressed as intensity normalized to KIF13B (****$P$ < 0.0001, unpaired $t$ test, n = 5 independent experiments). Bars show mean ± SEM.

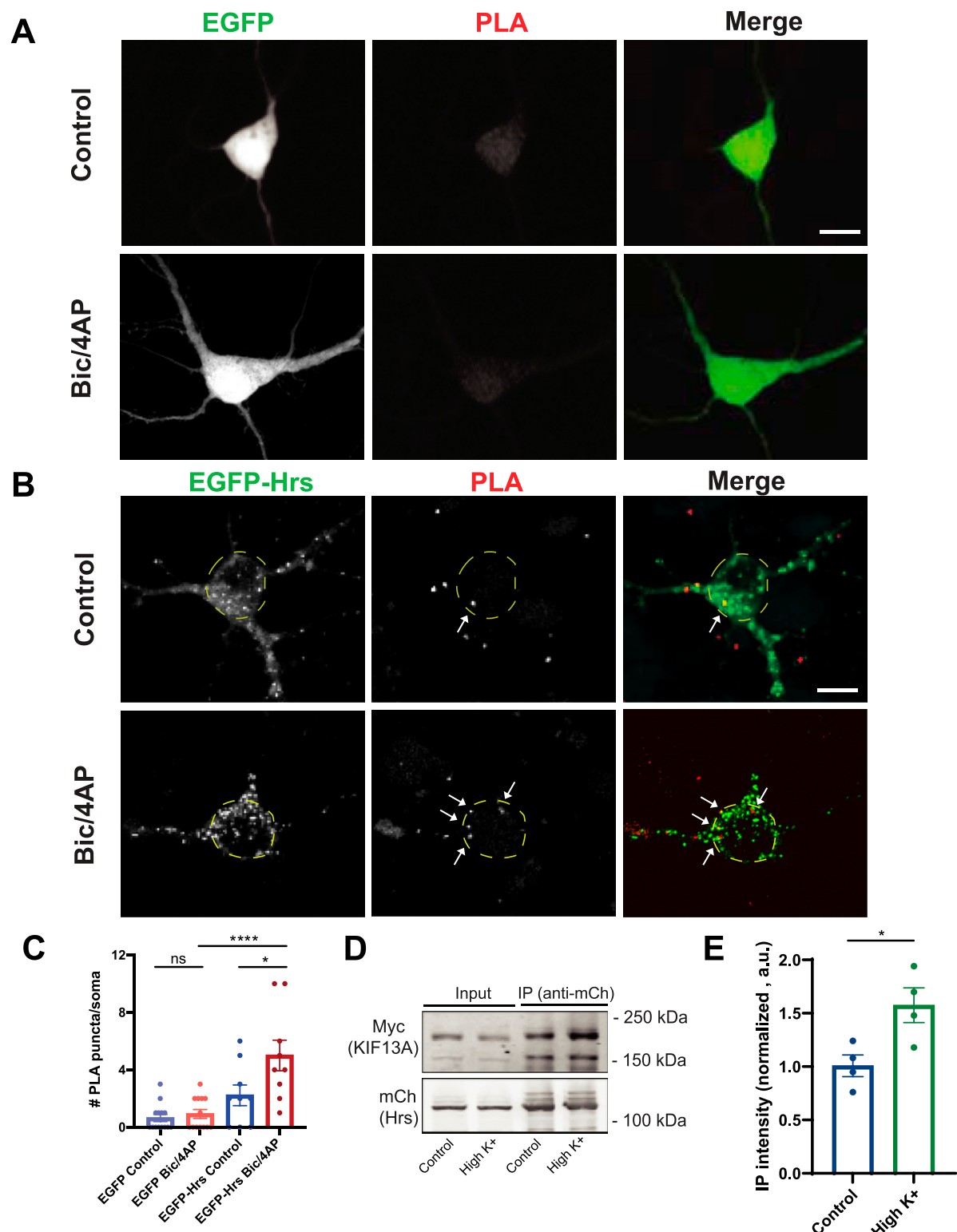

**Figure 5. Activity induces the interaction of Hrs with KIF13A.**

**(A, B)** Images of proximity ligation assay (PLA) in 13–15 day in vitro dissociated hippocampal neurons expressing EGFP (A) or EGFP-Hrs (B) under control or 2 h Bic/4AP conditions. PLA puncta (red), representing EGFP-Hrs/KIF13A interactions, were quantified in the soma (dashed yellow ovals). Size bar, 10 μm. **(C)** PLA puncta number per soma (*P = 0.0129, ****P < 0.0001, one-way ANOVA with Sidak's multiple comparisons test; n = 13 [EGFP control], 9 [EGFP-Hrs] condition, ≥3 independent experiments). **(D)** Immunoblot of lysates from N2a cells coexpressing RFP-Hrs and KIF13A-FRB-Myc, treated with control or high K+ Tyrodes solution for 2 h, immunoprecipitated (IP) with mCherry antibody and probed with Myc or mCherry antibodies. **(E)** Quantitative analysis of co-immunoprecipitated KIF13A, normalized to immunoprecipitated RFP-Hrs and expressed as intensity normalized to control condition (*P = 0.0249, unpaired t test, four independent experiments). Bars show mean ± SEM.

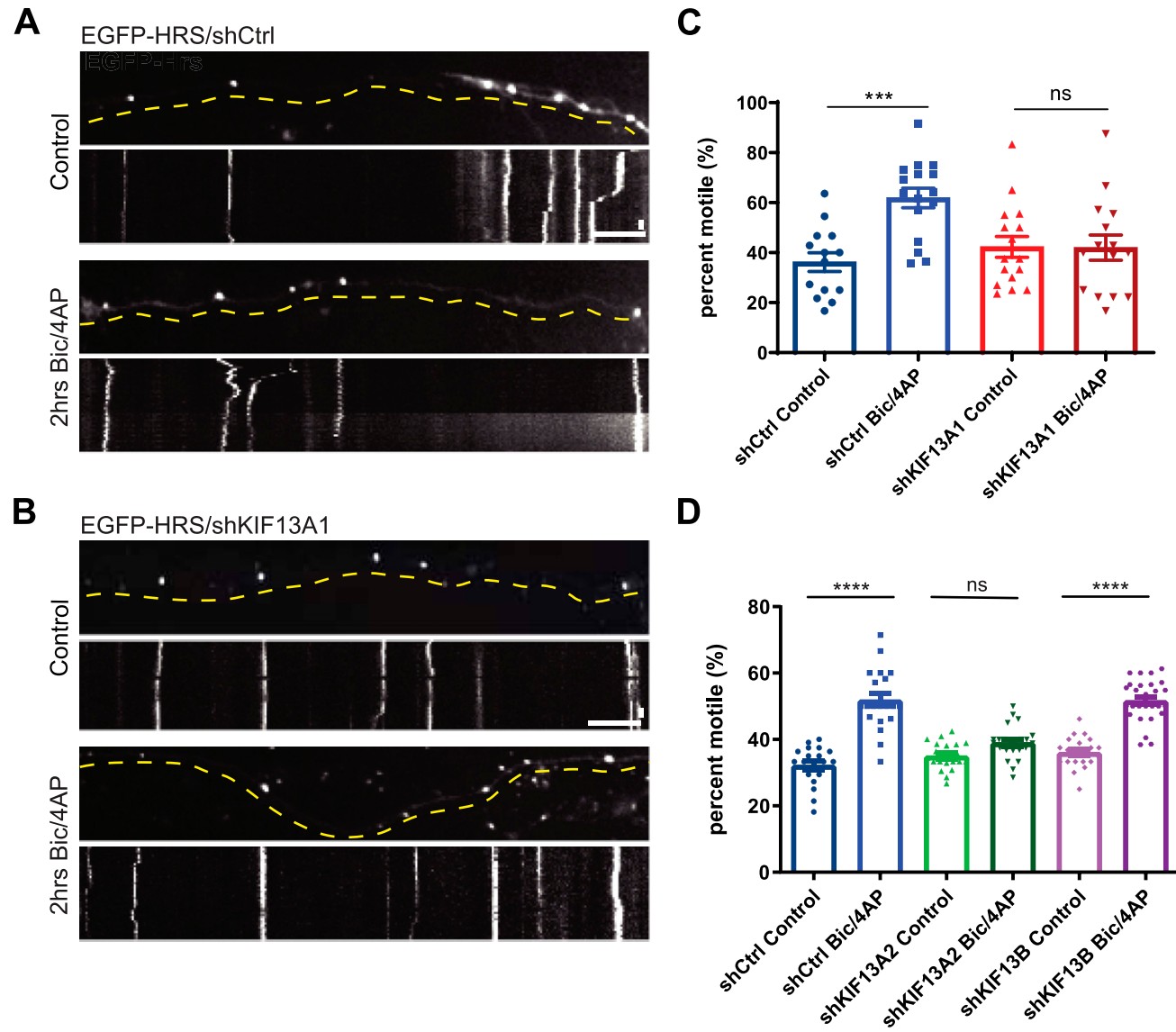

**Figure 6. Knockdown of KIF13A inhibits activity-dependent motility of Hrs.**

**(A)** Images and corresponding kymographs of EGFP-Hrs in microfluidically isolated axons expressing shCtrl under control conditions (upper panels) and after 2 h treatment with Bic/4AP (lower panels) (see Videos 7 and 8). **(B)** Images and corresponding kymographs of EGFP-Hrs in microfluidically isolated axons expressing shKIF13A1 under control conditions (upper panels) and after 2 h treatment with Bic/4AP (lower panels) (see Videos 9 and 10). Horizontal size bar = 10 $\mu m$, vertical size bar = 25 s. One frame taken every 5 s (0.2 fps). Dashed yellow lines highlight axons. **(C)** Percentage of motile Hrs puncta in axons (***$P$ = 0.0004, ns $P$ = 0.9985, one-way ANOVA with Sidak's multiple comparisons test; ≥3 separate experiments, n = 14 [shCtrl, control], 16 [shCtrl, 2 h Bic/4AP], 16 [shKIF13A1, control], and 15 [shKIF13A1, 2 h Bic/4AP] videos/condition). **(D)** Percentage of motile Hrs puncta in axons (****$P$ < 0.0001, ns $P$ = 0.1483, one-way ANOVA with Sidak's multiple comparisons test; three separate experiments, n ≥ 20 videos/condition). Bars show mean ± SEM.

they undergo multiple cycles of exo/endocytosis (Truckenbrodt et al, 2018), necessitating their targeting for degradation.

In addition to characterizing the dynamic behavior of Hrs+ transport vesicles, this study investigates the mechanism of Hrs targeting to these vesicles. As in non-neuronal cells (Komada & Soriano, 1999; Gaullier et al, 2000), we find that Hrs association with vesicles is dependent upon interactions between its FYVE domain and the endosomal lipid PI3P (Fig S5). Blocking PI3P synthesis with SAR405 or expressing the Hrs FYVE domain-inactivating mutant R183A both lead to a significant reduction of axonal Hrs vesicles (Fig S5). However, these manipulations do not completely abolish

Hrs' punctate expression in axons, suggesting that other domains also regulate its vesicle association. One possibility is the C-terminal coiled-coil domain, required for Hrs membrane targeting in non-neuronal cells (Raiborg et al, 2001). It is interesting to note that EEA1, which targets to early endosome membranes via its FYVE domain, does not localize to axons (Gaullier et al, 2000; Lawe et al, 2000; Wilson et al, 2000), indicating that additional domains and protein interactions may be required for axonal targeting of FYVE domain-containing proteins.

Our work has also elucidated a mechanism of activity-dependent transport of Hrs+ vesicles via KIF13A. Surprisingly, although both

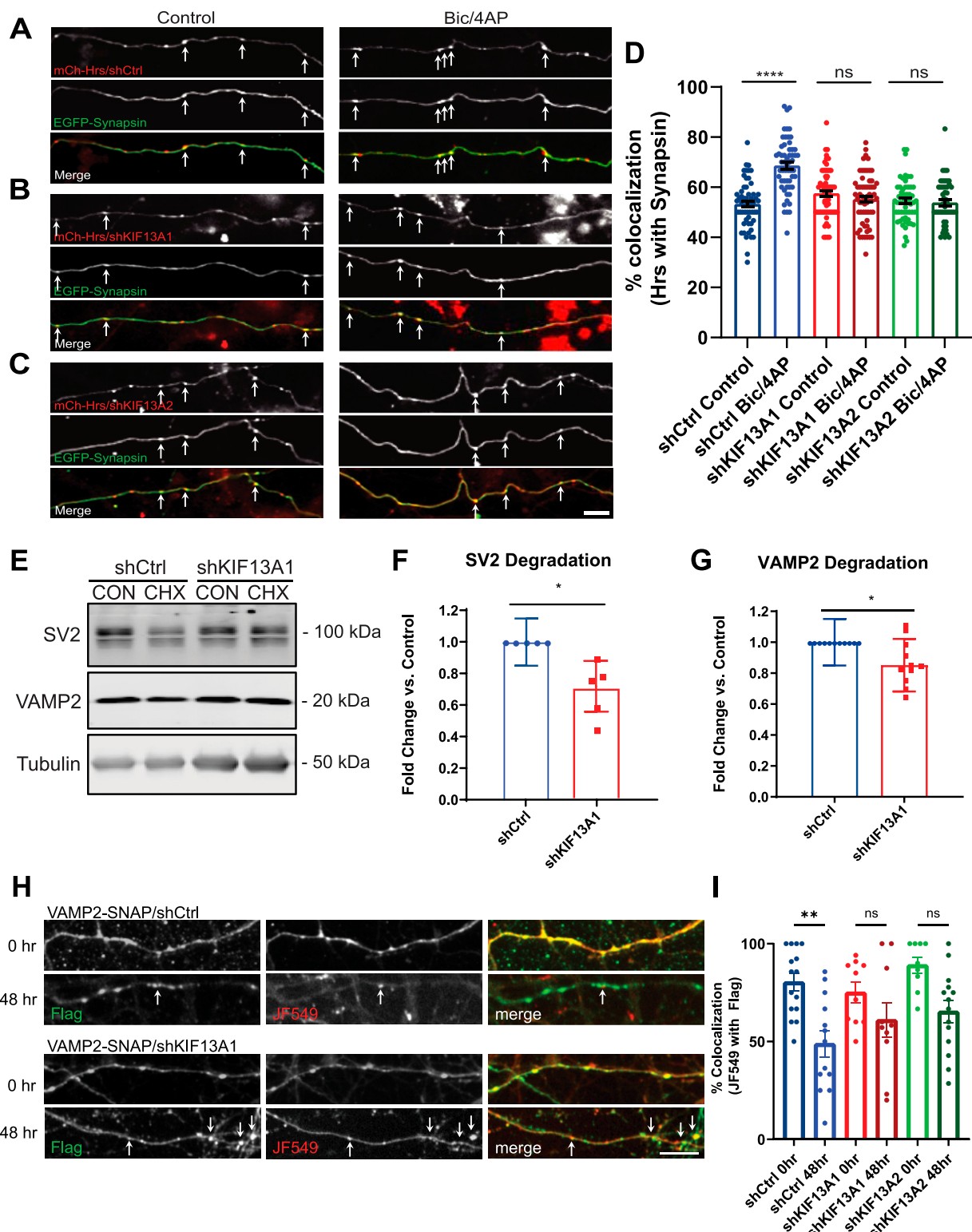

**Figure 7. Knockdown of KIF13A impairs activity-dependent transport of Hrs to synaptic vesicle (SV) pools and SV protein degradation.**
**(A, B, C)** Images of axons from 14 day in vitro (DIV) dissociated hippocampal neurons expressing mCh-Hrs/shCtrl (A), mCh-Hrs/shKIF13A1 (B), or mCh-Hrs/shKIF13A2 (C) together with EGFP-Synapsin, under control conditions (left panels) or after 20 h treatment with Bic/4AP (right panels). White arrows indicate sites of Hrs/Synapsin colocalization. Size bar, 10 μm. **(A, B, C, D)** Percentage of Hrs puncta colocalized with Synapsin under the conditions shown in (A, B, C) (****$P < 0.0001$, ns [KIF13A1] $P = 0.7548$, ns [KIF13A2] $P = 0.9997$, one-way ANOVA with Sidak's multiple comparisons test; three separate experiments, n ≥ 20 videos/condition). **(E)** Immunoblots of SV2 and VAMP2, with corresponding tubulin loading controls, from lysates of 14 DIV hippocampal neurons expressing shCtrl or shKIF13A1 and treated for 24 h with DMSO control

KIF13A and KIF13B mediate transport of Rab5 (Fig S7; [Bentley et al, 2015]), we find that only KIF13A transports Hrs (Fig 4). This specificity is confirmed by coimmunoprecipitation assays in which Hrs precipitates KIF13A more robustly than KIF13B, bolstering the concept that Hrs+ vesicles comprise a specialized subset of early endosomes with unique transport properties. KIF13A and KIF13B are highly homologous proteins that share multiple cargoes, including early and recycling endosome-associated proteins Rab5, Rab10, and transferrin receptor (Jenkins et al, 2012; Bentley et al, 2015; Etoh & Fukuda, 2019). These kinesins also have distinct cargoes, for example, low-density lipoprotein receptor for KIF13B and serotonin type 1a receptor for KIF13A (Jenkins et al, 2012; Zhou et al, 2013), potentially reflecting differences in their subcellular localizations in neurons. However, current reports are conflicting, with one study showing that KIF13A localizes to axons and dendrites, whereas KIF13B is primarily dendritic (Jenkins et al, 2012), and another showing that KIF13B has an axonal bias, whereas KIF13A is exclusively dendritic (Yang et al, 2019). In our experiments, we find that knockdown of KIF13A, but not KIF13B, prevents the activity-induced motility of Hrs in axons (Fig 6) as well as its delivery to SV pools (Fig 7), indicating that KIF13A alone has a role in the axonal transport of Hrs. Interestingly, KIF13A was recently found to mediate the delivery of AMPA-type glutamate receptors to the postsynaptic membrane upon induction of long-term potentiation, but not under baseline conditions (Gutierrez et al, 2021). These findings, like ours, indicate that KIF13A has a specific role in activity-dependent cargo transport in glutamatergic neurons.

Our findings highlight the importance of KIF13A-mediated transport in SV protein degradation, which is impaired by KIF13A knockdown under basal conditions (Fig 7). These data suggest that activity-dependent transport of Hrs occurs even under conditions of lower neuronal firing, but is likely difficult for us to capture during our brief imaging sessions. It is also possible that disruption of KIF13A-dependent trafficking inhibits SV degradation via another mechanism (e.g., interfering with autophagy). Indeed, there is considerable cross-talk between the ESCRT and autophagy pathways, and ESCRT proteins are essential for phagophore closure (Takahashi et al, 2018) as well as autophagosome/MVB fusion to form amphisomes (Ganesan & Cai, 2021). Future studies are needed to disentangle the roles of the ESCRT and autophagy pathways in SV protein degradation, as well as that of KIF13A in regulating transport of these degradative components. Another open question is which kinesins are required for the transport of Hrs and STAM1 under basal conditions, as KIF13A knockdown does not seem to impair this process. Several kinesins in addition to KIF13A and KIF13B, including KIF5A/B/C, KIFC1, and KIF16B, have been implicated in endosomal trafficking (Bonifacino & Neefjes, 2017; Li et al, 2020; Villari et al, 2020) and may also have roles in the axonal transport of ESCRT-0 proteins. Clearly, additional work is needed to identify other cargoes of KIF13A, as well

as the specific motor proteins responsible for anterograde and retrograde transport of Hrs+ vesicles.

The ESCRT pathway plays a vital role in maintaining neuronal health and is intimately linked to neurodegenerative disease etiology. Here, we have illuminated a novel mechanism for spatiotemporal regulation of the ESCRT pathway in neurons, namely, the activity-induced transport of Hrs to presynaptic SV pools by the kinesin motor protein KIF13A. These findings broaden our understanding of how protein degradation is regulated in neurons, and shed light on how disruption of this degradative pathway may contribute to the etiology of neurodegenerative disease.

# Materials and Methods

## Cell culture and transduction/transfection

Hippocampal neurons were prepared from E18 Sprague Dawley rat embryos of both sexes using a modified Banker culture protocol (Banker & Goslin, 1998; Kaech & Banker, 2006), as previously described (Sheehan et al, 2016). Neuro2a (N2a) neuroblastoma cells (ATCC CCL-131) and HEK293T cells (Sigma-Aldrich) were cultured in DMEM-GlutaMAX (Thermo Fisher Scientific/Invitrogen) with 10% FBS (Atlanta Biological) and Antibiotic-Antimycotic (Thermo Fisher Scientific/Invitrogen) and kept at 37°C in 5% $CO_2$. Microfluidic molds and chambers were prepared as described in Birdsall et al (2019). Chambers were prepared by pouring ~30 ml of PDMS mixture (QSil 216) into molds, then baking at 70°C overnight. Cell health in microfluidic chambers was evaluated using NucRed Dead 647 reagent to identify cells with compromised membranes along with membrane-permeant NucBlue Live ReadyProbes Reagent (both Thermo Fisher Scientific) to evaluate the total number of cells. Neurons were treated with these reagents for 15 min then immediately imaged as described (see Live Neuron Imaging). For neuronal transduction, lentivirus was produced using HEK293T cells as previously described (Sheehan et al, 2016; Birdsall et al, 2019). Neurons were transduced with 50–150 μl of virus on 7 DIV, and imaged or fixed on 13–15 DIV. For microfluidic chambers, ~50% of the media from the wells to be infected was removed and placed in adjacent wells, then virus was infected into the reduced-volume wells to contain it within those compartments (Fig S2) (Birdsall et al, 2019). Neurons were transfected as described in Sheehan et al (2016) using Lipofectamine 2000 (Thermo Fisher Scientific/Invitrogen) on nine DIV.

## Animals

$Hgs^{tn}$ mice were described previously (Watson et al, 2015). C57BL/6J (wild-type) mice were obtained from The Jackson Laboratory, $Hgs^{fl}$

---

(CON) or cycloheximide (CHX). **(F, G)** Quantification of the fold-change in SV2 (F) and VAMP2 (G) degradation under these conditions, calculated as previously described (Sheehan et al, 2016) (*$P$ = 0.0191 [SV2], *$P$ = 0.0161 [VAMP2], unpaired $t$ test; ≥3 separate experiments, n = 5 [SV2], 11 [VAMP2] cell lysates/condition). All scatter plots show mean ± SEM. **(H)** Images of axons from 13 to 15 DIV dissociated hippocampal neurons expressing VAMP2-SNAP/shCtrl, VAMP2-SNAP/shKIF13A1, or VAMP2/shKIF13A2, labeled with Janelia Fluor 549, then fixed and immunostained with Flag antibodies 0 or 48 h post-labeling. Yellow arrows indicate sites of JF549/Flag colocalization. **(I)** Percentage of total (Flag immunostained) VAMP2-SNAP with JF549 labeling under the conditions shown in H (**$P$ = 0.0027, ns [KIF13A1] $P$ = 0.6268, ns [KIF13A2] $P$ = 0.0567, two-way ANOVA with Tukey's multiple comparisons test, n = 15 [shCtrl, 0 h], 13 [shCtrl, 48 h], 10 [shKIF13A1, 0 and 48 h], 9 [shKIF13A2, 0 h], and 14 [shKIF13A2, 48 h] fields of view/condition).

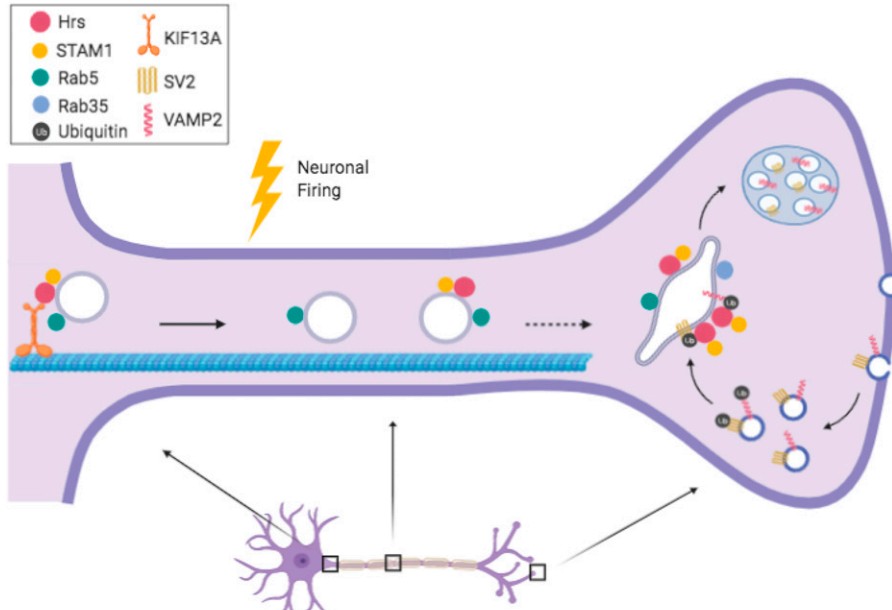

**Figure 8. Model for how neuronal activity stimulates the transport of Hrs to synaptic vesicle pools.** Neuronal firing increases the association of Hrs with plus-end directed kinesin motor protein KIF13A, stimulating the anterograde transport of these vesicles and their delivery to synaptic vesicle pools. Hrs+ vesicles also transport STAM1 and are a subset of Rab5+ early endosomes.

mice were obtained from the EMMA repository (strain B6Dnk;B6N-Hgs$^{tm1a(EUCOMM)Wtsi}$/Ieg) and bred with P0-Cre mice (The Jackson Laboratory, strain B6N.FVB-Tg(Mpz-cre)26Mes/J) for specific deletion of Hrs in Schwann cells. Animals were maintained at the University of Alabama at Birmingham, which is fully accredited by the Association for Assessment and Accreditation of Laboratory Animal Care International. All research was approved by the UAB IACUC committee and complied with the United States Animal Welfare Act and other federal statutes and regulations relating to animals and experiments involving animals and adhered to principles stated in the Guide for the Care and Use of Laboratory Animals, United States National Research Council.

## DNA constructs

pVenus-2xFYVE was a gift from F. Polleux and was subcloned into pFUGWm vector (Waites et al, 2013) at MfeI/NheI sites; pFU-RFP-Hrs-W, pFU-STAM1-mCh-W, pFU-EGFP/mCh-Rab35, pFU-EGFP/mCh-Rab5a, and pFU-VAMP2-Flag-SNAPtag were described in Sheehan et al (2016); pFU-EGFP-Synapsin-W was described in Leal-Ortiz et al (2008); and pBa-EGFP-flag-BicD2594-FKBP (plasmid #63569; http://n2t.net/addgene:63569; RRID:Addgene_63569; Addgene), pBa-FRB-3myc-KIF13A tail 361–1749 (plasmid #64288; http://n2t.net/addgene:64288; RRID:Addgene_64288; Addgene), and pBa-FRB-3myc-KIF13B tail 442–1826 (plasmid #64289; http://n2t.net/addgene:64289; RRID:Addgene_64289; Addgene) were gifts from Gary Banker and Marvin Bentley (Jenkins et al, 2012; Bentley & Banker, 2015; Bentley et al, 2015). Full-length human HRS (NCBI accession #D84064.1) and human CHMP4B (NCBI accession #NM_176812.5) were synthesized at Genewiz and cloned into pFUGWm vector at BsrGI/XhoI sites to create pFU-EGFP-Hrs-W and pFU-EGFP-Chmp4b-W. For Hrs, EGFP was replaced with Halotag/mCherry at the AgeI/BsrGI sites to create pFU-Halo-Hrs-W/pFU-mCherry-Hrs. To create pFU-EGFP-

R183A Hrs-W, an 868 nucleotides fragment of Hrs containing the R183A mutation was synthesized at Genewiz and subcloned into the BsrGI/PshAI sites. Full-length [Chmp4b] shRNA scrambled control and KIF13A knockdown constructs (shCtrl, shKIF13A1 and A2, KIF13B) were purchased from Origene (Cat. no. TL706020). The KIF13B knockdown construct was a gift from Dr. Richard Vallee (Dept of Pathology & Cell Biology, CUMC). After testing knockdown efficacy, the 29mer or 21mer shRNA (lower-case letters) and surrounding loop sequences (capital letters) (shCtrl: CGGATCGgcactaccagag-ctaactcagatagtactTCAAGAGagtactatctgagttagctctggtagtgcTTTTT; shKIF13A1:CGGATCGagccagacctctatgatagcaaccatcagTCAAGAGctgatggtt-gctatcatagaggtctggct TTTTT;shKIF13A2:CGGATCGtacgaggagacactgtc-cacgctaagataTCAAGAGtatcttagcgtggacagtgtctcctcgtaTTTTT;shKIF13B: GATCCctgtggaagtaattgctgcaaTCAAGAGttgcagcaattacttccacagTTTTT) were then re-synthesized and subcloned into pFUGW U6 vector (Waites et al, 2013) at EcoRI/PacI sites. shRNAs were moved into pFU-EGFP-Hrs-W and pFU-VAMP2-Flag-SNAPtag vectors using MluI/PacI digestion.

## Antibodies and chemical reagents

The following antibodies were used: VAMP2 rabbit antibody (#104202; Synaptic System), SV2 mouse antibody (Developmental Studies Hybridoma Bank), Flag M2 mouse antibody (#1804; Sigma-Aldrich), tubulin mouse antibody (#t9026; Sigma-Aldrich), mCherry rabbit antibody (#5993; BioVision), Myc mouse antibody (9E10; Santa Cruz), Hrs rabbit antibody (D7T5N; Cell Signaling), Hrs mouse antibody (ab56468; Abcam), GFP mouse antibody (#11814460001; Roche), KIF13A rabbit antibody (PA5-30874; Thermo Fisher Scientific), KIF13B mouse antibody (6E11; Santa Cruz), neurofilament and β-actin mouse antibodies (2H3 and 8-7A5; Developmental Studies Hybridoma Bank). Pharmacological agents were used in the following concentrations and time courses: cycloheximide

(Calbiochem, 0.2 μg/μl, 24 h), bicuculline (Sigma-Aldrich, 40 μM, 5 min-2h or 20 h), 4-aminopyridine (Tocris Bioscience, 50 μM, 5 min-2h or 20 h), BDNF (R&D Systems, 50 ng/ml, 1 h), TTX (Tocris Bioscience, 0.5 μM, 2 h), Rapamycin analogue linker drug (AP21967; Clontech, 100 nM, 3 h), SAR405 (Millipore Sigma-Aldrich, 1 μM, 24 h), and Janelia Fluor 549 SNAPtag ligand (Janelia Research Campus, 200 nM, 30 min). Unless otherwise indicated, all other chemicals were purchased from Sigma-Aldrich.

### Electron micrograph analysis of endosomal structures

Electron microscopy of hippocampal synapses was performed by Glebov and Burrone as described (Glebov et al, 2016). Presynapses were defined visually by the presence of SV pools. Endosomal structures were defined as single-membraned structures with a diameter >80 nm. Multilamellar structures were defined a structure with more than one visible membrane. Synaptic area and endosomal/multilamellar structure size were measured using FIJI.

### Correlative light and electron microscopy

For correlative fluorescence and electron microscopy experiments, mouse hippocampal neurons were cultured on sapphire disks (#616-100; Technotrade) that were carbon-coated in a grid pattern (using finder grid masks, #16770162; Leica microsystems) to locate the region of interest by electron microscopy. After 1 mg/ml poly-L-lysine (#P2636; Sigma-Aldrich) coating overnight, neurons were seeded at densities of $2.5 \times 10^2$ cell/mm$^2$ in 12-well tissue-culture treated plates (#3512; Corning) and cultured in NM0 medium (Neurobasal medium (#21103049; Gibco)) supplemented with 2% B27 plus (# A3582801; Life Technologies) and 1% GlutaMAX (#35050061; Thermo Fisher Scientific). STAM-Halo, HRS-Halo, and vGlut1-pHluorin were lentivirally transduced at six DIV, and experiments performed at 14–15 DIV. To induce activity, 40 μM bicuculline (Bic, #0130/50; Tocris) and 50 μM 4-aminopyridine (4AP, #940; Tocris) were added to the culture media at 37°C. After 2 h of incubation, half of the medium was changed to NM0 with 2 mM Jenelia-Halo-549 nm (#SB-Jenelia-Halo-549; Lavis Lab) and incubated for an additional 30 min at 37°C. Neurons on the sapphire disks were washed 3× with PBS, fixed with 4% paraformaldehyde (#15714; EMS) and 4% sucrose (#S0389; Sigma-Aldrich) in PBS for 20 min at room temperature, washed again 3× with PBS, and mounted on a confocal scanning microscope (LSM880; Carl Zeiss) using a chamber (#MS-508S; ALA Science). Z-series of fluorescence images were acquired using a 40× objective lens at 2,048 × 2,048 pixel resolution. Differential interface contrast images of the carbon grid pattern and neurons on the sapphire disks were also acquired to locate the cells in later steps.

After fluorescence imaging, neurons were prepared for electron microscopy. Here, sapphire disks were placed into 12-well plate containing 1 ml of 2% glutaraldehyde (#16530; EMS) and 1 mM CaCl$_2$ (# 63535; Sigma-Aldrich) in 0.1 M sodium cacodylate (#C0250; Sigma-Aldrich), pH 7.4, and incubated for 1 h on ice. Subsequently, the sapphire disks were washed with 1 ml of chilled 1M cacodylate buffer 3× for 5 min each. Neurons were then postfixed for 1 h on ice with 2 ml of 1% osmium tetroxide (#RT19134; EMS) and 1% potassium ferrocyanide (#P3289; Sigma-Aldrich) in 0.1 M sodium cacodylate,

pH 7.4. After washing 3× with 2 ml of water for 5 min, neurons were stained with 1 ml of 2% uranyl acetate (#21447-25; Polysciences) in water for 30 min at room temperature and then briefly washed with 50% ethanol (#111000200; Pharmco) before dehydration in a graded ethanol series (50%, 70%, and 90% for 5 min each, and 100% for 5 min 3×). After dehydration, sapphire disks were embedded into 100% epon-araldite resin (#18005, 18060, 18022; Ted Pella) with the following schedule and conditions: overnight at 4°C, 6 h at room temperature (solution exchange every 2 h), and 48 h at 60°C.

For electron microscopy imaging, regions of interest were located based on the bright field images. When a sapphire disk is removed, neurons and carbon-coating remain in the plastic block, which was trimmed to the regions of interest and sectioned with a diamond knife using an ultramicrotome (UC7; Leica). Approximately 40 consecutive sections (80 nm each) were collected onto the pioloform-coated grids and imaged on a Phillips transmission electron microscope (CM120) equipped with a digital camera (XR80; AMT). The fluorescence and electron micrographs were roughly aligned based on size of the pixels and magnification, and the carbon-coated grid patterns. The alignment was slightly adjusted based on the visible morphological features.

### Live imaging

Neurons were imaged at 12–15 DIV after being transduced on seven DIV or transfected at nine DIV. Imaging was conducted in pH-maintaining Air Media, consisting of 250 ml L15 Media (Thermo Fisher Scientific/Invitrogen) and 225 ml Hanks Balanced Salt Solution (Sigma-Aldrich), supplemented with 55 mM Hepes, 2% B27 (Thermo Fisher Scientific), 0.5 mM GlutaMAX (Thermo Fisher Scientific), 0.1% Pen/Strep, 0.5% glucose, and 2.5% BME. Neurons were imaged at 37°C with a 40× oil-immersion objective (NA 1.3; Neofluar) on an epifluorescence microscope (Axio Observer Z1; Zeiss) with Colibri LED light source, EMCCD camera (Hamamatsu) and Zen 2012 (blue edition) software. One frame was taken every 5 s, for a total of 50 frames. Images were obtained and processed using Zen Blue 2.1 software. Axons were identified via location and morphological criteria. Image processing was performed using FIJI (Image-J).

### Live imaging analysis

Motility of axonal puncta was analyzed using FIJI/ImageJ2 "Manual Tracker" plugin. Raw data for the movement of each puncta over each video frame were imported into Matlab R2018b and motility, directionality, and speed were calculated using a custom program. Only puncta ≥0.5 and <1.5 μm in diameter were analyzed. Directionality of each puncta was based on net displacement over the course of the video (Fig S3I), and was categorized as follows: ≥4 μm away from the cell body = anterograde, ≥4 μm towards the cell body = retrograde, ≥4 μm total, but <4 μm in one direction = bidirectional, <4 μm = stationary. Axon length was measured in FIJI.

### Immunofluorescence microscopy

Neurons or N2a cells were immunostained as described previously (Leal-Ortiz et al, 2008; Sheehan et al, 2016). Coverslips were either mounted with DAPI VectaShield (Vector Laboratories) and sealed

with clear nail polish, or with Aqua-Poly/Mount (Polysciences) and dried overnight in the dark. Images were acquired on the epifluorescence microscope described above (Axio Observer Z1; Zeiss), or on a Zeiss LSM 800 confocal microscope using a 63× objective (Plan-Apochromat, NA 1.4). Images were obtained and processed using Zen Blue 2.1 software. Image processing was performed using FIJI (Image-J).

### In vitro KIF13A/KIF13B transport assay

N2a cells were transfected at 50–70% confluence with BicD2-GFP-FKBP, KIF13A-FRB-Myc or KIF13B-FRB-Myc, and either mCh, mCh-Rab5, or RFP-Hrs, using Lipofectamine 3000 (Thermo Fisher Scientific/Invitrogen) according to manufacturer's instructions. After 48 h, rapamycin analogue (AP21967) or EtOH vehicle control were added to relevant samples as described in Bentley and Banker (2015).

### SNAPtag labeling

Hippocampal neurons were lentivirally transduced with VAMP2-SNAP/shCtrl or VAMP2-SNAP/shKIF13A1 on 7 DIV. On 14 DIV, cell-permeant Janelia Fluor 549-labeled SNAPtag ligand was diluted to a final concentration of 200 nM in Neurobasal medium. Neurons were incubated for 30 min at 37°C, gently washed three times with fresh Neurobasal medium, and returned to their original conditioned medium. At 0 and 48 h after labeling, neurons were fixed, immunostained with anti-Flag antibody, and imaged as described in the Immunofluorescence Microscopy section above.

### Colocalization analysis

Colocalization analysis was performed with Fiji/ImageJ, using the Colocalization plugin (Pierre Bourdoncle) together with the Analyze Particles function. Briefly, puncta above a threshold value (typically <5% in threshold window) were determined empirically for each channel, ROIs (7 × 7, oval) were drawn around puncta in the target channel (e.g., Hrs), and the Analyze Particles function was used to count total versus colocalized puncta. For VAMP2-SNAP colocalization analysis, the selected puncta were recentered around the highest intensity pixel using the Time Series Analyzer V3 plugin (Balaji), and colocalization analysis was performed semi-automatically using the ComDet v.0.5.5 plugin (Katrukha). The threshold for detection in this program was determined empirically for each channel (typically 3–10). A colocalization event was defined as one in which the maximum intensity pixels of each channel (Flag, JF549) were no more than three pixels apart from one another (the average size of the Flag puncta); thus, any overlap of Flag/JF549 puncta counted as colocalization. The percent colocalization was calculated within the plugin.

### PLA

PLA was performed in hippocampal neurons according to manufacturer's instructions (Duolink; Sigma-Aldrich). Until the PLA probe incubation step, all manipulations were performed as for immunofluorescence microscopy. PLA probes were diluted in blocking solution, and the primary antibody pairs used were anti-KIF13A (Rabbit, 3 µg/ml) and anti-GFP (Mouse, 1 µg/ml). All protocol steps were performed at 37°C in a humidified chamber, except for the washing steps. Coverslips were then mounted using Duolink In situ Mounting Media with DAPI.

### Immunoprecipitation and Western blot

For immunoprecipitation assays, N2a cells were transfected at 50–70% confluence using Lipofectamine 3000 (Thermo Fisher Scientific/Invitrogen) according to manufacturer's instructions. Cells were collected 48 h later in lysis buffer (50 mm Tris-Base, 150 mm NaCl, 1% Triton X-100, 0.5% deoxycholic acid) with protease inhibitor mixture (Roche) and clarified by centrifugation at high speed (20,000 rcf). Resulting supernatant was incubated with Dynabeads (Thermo Fisher Scientific/Invitrogen) coupled to anti-mCherry antibodies (polyclonal; BioVision). Lysates and beads were incubated at 4°C under constant rotation for 2 h. Beads were washed 2–3× with PBS containing 0.05% Triton (PBST) and then once with PBS. Bound proteins were eluted using sample buffer (Bio-Rad) and subject to SDS–PAGE immunoblotting as described following. For other immunoblotting experiments from cultured neurons, lysates were collected directly in 2× SDS sample buffer (Bio-Rad). Samples were subjected to SDS–PAGE, transferred to nitrocellulose membranes, and probed with primary antibody in 5% BSA/PBS + 0.05% Tween 20 overnight at 4°C, followed by DyLight 680 or 800 anti-rabbit, anti-mouse, or anti-goat secondary antibodies (Thermo Fisher Scientific) for 1 h. Membranes were imaged using an Odyssey Infrared Imager (model 9120; LI-COR Biosciences). Protein intensity was measured using the "Gels" function in FIJI. Protein collection and immunoblotting from $Hgs^{tn}$, $Hgs^{fl}$, and $P0Cre\text{-}Hgs^{fl}$ mice was performed as previously described (Watson et al, 2015).

### Statistical analyses

All statistical analyses were performed in GraphPad Prism 9. Statistics for each experiment are described in the Figure Legends. A D'Agostino–Pearson normality test was used to assess normality. An unpaired $t$ test or a Mann–Whitney U test was used to compare two datasets, and an ANOVA or a Kruskal–Wallis test was used to compare three or more datasets. For multiple comparisons, Dunnett's or Dunn's post hoc test was used to compare a control mean to other means, whereas Sidak's post hoc test was used to compare pairs of means. Statistical significance is noted as follows (and in Figure Legends): ns, $P > 0.05$; *$P \leq 0.05$; **$P \leq 0.01$; ***$P \leq 0.001$, ****$P \leq 0.0001$.

# Data Availability

The data that support the findings of this study are available from the corresponding author upon reasonable request.

# Supplementary Information

# Acknowledgements

We would like to thank Drs. Oleg Glebov and Juan Burrone (King's College, London, UK) for sharing EM micrographs of synapses from hippocampal neurons treated with TTX and gabazine, Dr. Ulrich Hengst (Columbia University) and Hengst lab members for help with microfluidic chambers, Dr. Kapil Ramachandran and Ramachandran lab members for help with SNAPtag labeling and for sharing Janelia Fluor SNAPtag reagents, Dr. Richard Vallee for the shKIF13B hairpin construct, and members of the Waites lab for their scientific input. This work was supported by National Institutes of Health grants 2R01NS080967 to CL Waites and S Watanabe, 1RF1AG069941 to CL Waites, 5T32NS064928 to V Birdsall, National Science Foundation grant DGE-2036197 to K Kirwan, and Hirschl Research Scientist Award to CL Waites. The authors declare no competing financial interests.

## Author Contributions

V Birdsall: conceptualization, data curation, software, formal analysis, validation, investigation, visualization, methodology, and writing—original draft.
K Kirwan: data curation, software, formal analysis, validation, investigation, visualization, and methodology.
M Zhu: data curation, formal analysis, validation, investigation, visualization, and methodology.
Y Imoto: data curation, formal analysis, investigation, visualization, and methodology.
SM Wilson: resources, validation, and investigation.
S Watanabe: data curation, supervision, investigation, methodology, and project administration.
CL Waites: conceptualization, resources, data curation, supervision, funding acquisition, methodology, project administration, and writing—original draft, review, and editing.

## Conflict of Interest Statement

The authors declare that they have no conflict of interest.

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
