## [Reviewer comments · Life Science Alliance]

Life Science Alliance

Axonal transport of Hrs is activity-dependent and facilitates synaptic vesicle protein degradation

Veronica Birdsall, Konner Kirwan, Mei Zhu, Yuuta Imoto, Scott Wilson, Shigeki Watanabe, and Clarissa Waites

DOI: <https://doi.org/10.26508/lsa.202000745>

Corresponding author(s): Clarissa Waites, Columbia University Medical Center

Review Timeline:

Submission Date:	2020-04-19
Editorial Decision:	2020-06-04
Revision Received:	2022-04-13
Editorial Decision:	2022-05-16
Revision Received:	2022-05-18
Accepted:	2022-05-19

Scientific Editor: Novella Guidi

Transaction Report:

June 4, 2020

Re: Life Science Alliance manuscript #LSA-2020-00745-T

Dr. Clarissa L Waites
Columbia University
Pathology and Cell Biology
630 W. 168th St.
Black Building 1210B
New York, NY 10032

Dear Dr. Waites,

Thank you for submitting your manuscript entitled "KIF13A mediates the activity-dependent transport of ESCRT-0 proteins in axons" to Life Science Alliance. The manuscript was assessed by expert reviewers, whose comments are appended to this letter.

As you will see, while the reviewers appreciate the interesting findings presented in your manuscript, they also raise numerous concerns. We do not rule out however that you might be able to address a majority of these points, and given the overall high level of interest in your study, we would still like to invite you to submit a revised version.

When submitting the revision, please include a letter addressing the reviewers' comments point by point. While each of the specific points that were raised should be addressed, we do feel that the suggestion by Reviewer 2 to perform additional domain analyses of Hrs is outside the scope of the current study and does not need to be addressed during the revision. We would be happy to discuss the other individual revision points further with you should this be helpful.

In our view these revisions should typically be achievable in around 3 months. However, we are aware that many laboratories cannot function fully during the current COVID-19/SARS-CoV-2 pandemic and therefore encourage you to take the time necessary to revise the manuscript to the extent requested above. We will extend our 'scooping protection policy' to the full revision period required. If you do see another paper with related content published elsewhere, nonetheless contact me immediately so that we can discuss the best way to proceed.

To upload the revised version of your manuscript, please log in to your account: <https://lsa.msubmit.net/cgi-bin/main.plex>. You will be guided to complete the submission of your revised manuscript and to fill in all necessary information. Please get in touch in case you do not know or remember your login name.

Please note that papers are generally considered through only one revision cycle, so strong support from the referees on the revised version is needed for acceptance.

Thank you for this interesting contribution to Life Science Alliance. We are looking forward to receiving your revised manuscript.

Sincerely,

Reilly Lorenz
Editorial Office Life Science Alliance
Meyerohofstr. 1
69117 Heidelberg, Germany
t +49 6221 8891 414
e contact@life-science-alliance.org
www.life-science-alliance.org

B. MANUSCRIPT ORGANIZATION AND FORMATTING:

Reviewer #1 (Comments to the Authors (Required)):

Summary:

The authors characterize how neuronal activity regulates the Kif13a-dependent trafficking of the two ESCRT-0 proteins HRS and Stam1. Previous work by these authors showed that synaptic vesicle (SV) proteins are degraded in an ESCRT dependent manner. This SV protein degradation was also shown to be activity-dependent. These data led the authors to further characterize how activity regulates the transport of ESCRT-0 proteins. The authors find that anterograde and bidirectional trafficking of Hrs and Stam1 (as well as a subset of Rab5 endosomes) increases in an activity dependent manner and that this movement is dependent on the kinesin Kif13A. The paper is concluded with a model in which neuronal activity leads to increased anterograde trafficking of the ESCRT-0 complex to begin SV protein turnover.

Major points:

1) One widespread problem in the paper is the quality of the imaging data. The main concern of this reviewer is that many the puncta that are depicted in kymographs cannot be followed/tracked (e.g. Fig 2B+C, 2H (what is considered a Synapsin punctum is not obvious), Fig 3A and 5F). On top of this problem, the authors routinely find 40% motility in control settings where there are basically no moving puncta shown in the representative image (which begs the question, what movement is being tracked?). For examples see Fig2A, Fig 3A, Fig5E-i and Fig 7D+E. Lastly, the quality of the imaging data is also a problem in the colocalization analyses as the images shown appear saturated (Figure 6A and C).

2) In many places in the manuscript, statistics are handled inappropriately when data are reported as "n.s." or the claim is made that a parameter didn't change. In these cases, the actual p values always need to be given: after all, a change with a $p=0.051$ is still 20 times more likely to be an actual difference than an artifact. And, even when the p value is much higher, it does not demonstrate the absence of a difference, only that the difference could not be detected. One such place, early in the text, is in Figure 2F where retrograde transport is described as not increasing with activity. The % moving retrograde clearly has increased however, but we are not informed of the magnitude, nor the p value. This is an important point since the authors make much of the fact that anterograde, but not retrograde transport is enhanced by activity. A related concern is the distinction between statistical significance and biological significance. It is true that anterograde movement is increased, but this increase is very small in actual terms of the % of Hrs puncta, especially when compared to those puncta in non-processive bidirectional movement which accounts for almost all the decrease in the stationary pool. In figure 2G, it is again unclear how likely it is that there was an acute change in colocalization since there is no p value given.

3) The health of the neuronal cultures is assessed with vital dyes, but that is not a very sensitive method for judging neuronal health. Older microfluidic cultures can be on the decline and the authors should include some standard dissociated hippocampal cultures to determine if the motility parameters are comparable.

4)

Additional points:

1) Introduction

The premise of the work is that SV proteins need to be retrogradely transported to the soma for degradation and that ESCRT proteins need to be anterogradely transported in axons to synapses so that the initial steps for SV protein degradation can occur locally. The introduction asserts these premises largely without citations and indeed some of these premises are on shaky ground. There are lysosomes in axons, there is retrograde transport of SVs which could permit quality control in the soma without ESCRT in the periphery, and there is ESCRT present already in the periphery so that additional activity-mediated transport may not be essential. Truckenbrodt, for example, found that some older synaptotagmin was found in the soma, but also that inhibition of lysosomes caused a local increase in synaptotagmin, implying local degradation as well as retrograde transport.

2) Figure 1.

--Concerning the representative pictures in 1a and 1b, it would be nice if they showed representative images for the TTX and gabazine treatments (at least for the endosomes). It is my understanding that the pictures in 1A and 1B are just to show what they are scoring as endosomes and multilamellar structures.

--it would be preferable to show actual p-values rather than stars or ns (Fig 1c and 1D) even if the $P < 0.0001$ as written in the figure legend.

3) Figure 2.

--It is hard to see Hrs puncta in 2C and the still image does not quite match up with the kymograph. Also, please indicate the time span of the kymograph as well the acquisition rate (fps) of the time lapse (for all time lapse experiments).

--In figure 2h instead of performing colocalization of 2 overexpressed proteins, it would be preferable to determine the colocalization of the endogenous proteins. More specifics on how % time colocalized was determined would also be appreciated.

4) Figure S1.

- S1 A. Do the neurons seeded in the tripartite chamber efficiently form synapses at DIV13-15?

--S1 B, C. Please clarify how total number of neurons were counted in order to calculate % transduced cells (e.g. DIC, Hoechst). If dye was used, were glia eliminated from count?

-- S1 D. the data spread for the control is somewhat concerning because it looks bimodal and the n is pretty low. It is also concerning that it there appears there is no difference in motility between the TTX and Bic/4AP samples.

5) Figure 4.

--The authors should show a better representative picture for Hrs (Fig 4A).

6) Figure 7. I understand that SV2 is blurry on blots, but the Western blot does not instill confidence in the argument the authors are trying to make.

7) Minor point. The way the IP is described for Fig 6 needs to be toned down: "Consistent with our transport assays, we found that Hrs clearly precipitated KIF13A, but not KIF13B (Fig. 6E-F). STAM1 also had some ability to precipitate KIF13A, albeit variably, but did not precipitate KIF13B (Fig. 6E-F)." However, in 6E, there is a clear Myc band in the IP for both Stam and Hrs (albeit less than Kif13A but it is still very much visible to the reader).

Reviewer #2 (Comments to the Authors (Required)):

The authors present novel and intriguing data supporting a mechanism of activity-dependent trafficking of ESCRT-0 proteins by KIF13A. In particular, the authors demonstrate that populations of Hrs and STAM1 vesicles are stimulated to move upon acute and prolonged (2 hr) treatment with bicuculline and 4-aminopyridine and that this activity-dependent movement is repressed when the kinesin-3 KIF13A is knocked down. This study opens up a number of interesting questions regarding how this trafficking is accomplished and what its consequences are, not all of which are answered by the data presented. In particular, while the domain analysis of the Hrs FYVE domain is interesting in that it confirms that this domain is important for Hrs loading onto vesicles in the somatodendritic compartment of neurons, it does not provide insight into how Hrs is able to respond to neuronal activity. This study would be greatly strengthened if the authors could perform a domain analysis using other Hrs truncations to identify which domain is responsible for the activity-dependent motility and whether this Hrs domain or another directly interacts with KIF13A to initiate transport. Additionally, the authors state throughout the text that the activity-dependent motility of the ESCRT-0 proteins is important for transport to and degradation of SV pools; the data presented do not directly support this claim, however. Additional examination of Hrs/STAM1 activity-dependent re-localization (do they ever accumulate at SV pools?) and the ubiquitination state of SV proteins upon stimulation (providing a link to the statistically significant PR619 data) would strengthen this conclusion. Overall, this study contains exciting, varied techniques, is well written, and provides new insight into axonal trafficking of Hrs/STAM1. However, the authors overstate some of their findings and the study would require additional experiments to fully elucidate the mechanism and consequence of the observed activity-dependent ESCRT protein trafficking.

Given these concerns, this manuscript is currently not suitable for publication but could be acceptable if the concerns listed below are addressed.

The following specific major concerns should be addressed:

1. Page 5: "Under control conditions, we observed that >60% of Hrs puncta were stationary over the course of the ~4-minute imaging session (Fig. 2A, D, F). Acute treatment with bic/4AP led to a slight but significant increase in puncta motility (from ~38% to ~50% motile), with the effect growing stronger after 2h of treatment (~60% motile puncta; Fig. 2A-D, F)."
 - The observed increase in bidirectional motility doesn't inform the reader if the movement is ultimately processive. Net displacement measurements are needed to demonstrate processive anterograde trafficking. Additionally, the kymographs of Hrs motility in Figure 2B, C, and H do not support the idea that Hrs undergoes deliberate anterograde trafficking upon neuron firing. Additional analysis of the movement and whether it ultimately leads to co-localization with SV pools (as suggested below in point 2) is needed to support the authors conclusion that the Hrs trafficking is important for SV degradation.

2. Page 6: "While we did observe motile Hrs puncta colocalizing with Synapsin at presynaptic sites (Fig. 2G-H, Video 3), this behavior did not depend upon neuronal activity, as motile Hrs puncta spent ~35% of their time in contact with Synapsin regardless of activity condition."
 - This is concerning, as the proposed role of the trafficking is to promote activity-dependent degradation of SV pools. Are there more representative kymographs that display Hrs moving to a site of Synapsin and pausing? The authors' live imaging data could be mined to determine the percent of Hrs/STAM1 puncta that move to and pause at Synapsin sites. This type of analysis of ESCRT-0 protein movement could be very informative and support their conclusions.

3. Page 6: "However, Hrs/Synapsin colocalization was significantly increased following 2h of treatment with PR619, a deubiquitinating enzyme inhibitor that promotes SV protein ubiquitination (Fig. S1E-F), suggesting that Hrs interaction with presynaptic sites is ubiquitin-dependent."
 - This is very interesting, but as the increased co-localization was not observed with the bic-4AP treatment it opens up additional questions. Looking at VAMP2-ubiquitination (as in Supplemental Figure 1E) under various neuronal stimulation conditions could help link this finding to the idea that neuronal activity induces SV protein ubiquitination and subsequent ESCRT-dependent degradation.

4. Page 6: "Intriguingly, Hrs-positive structures identified in the axon appear to be degradative in nature, including an endosome and a multivesicular body (Fig. S2). These data, along with our previous findings (Sheehan et al. 2016), suggest that Hrs undergoes activity-dependent transport in axons and is associated with degradative structures."
 Page 11: "Moreover, our CLEM experiments demonstrate that Hrs+ vesicles are degradative structures (Fig. S2), though we cannot yet correlate their morphology with their axonal transport history to determine what stage of the degradative process they represent."
 - Based solely on the EM data, the authors cannot state that the Hrs+ structures are degradative. Follow-up experiments are needed to confirm this statement.

5. Page 8: " Interestingly, while the 2xFYVE domain exhibited highly punctate expression in axons, its dynamics were different from those of full-length Hrs and it did not demonstrate activity-dependent changes in motility (Fig. S4A-F), indicating that other domains of Hrs facilitate its activity-dependent transport."
 - This study would greatly benefit from additional Hrs domain analysis. If the authors could identify the domain that is responsible for the activity-dependent trafficking, and better yet link the domain to an association, either direct or indirect, with KIF13A, the conclusions of this paper would be more strongly supported.

6. Page 12: "Finally, our work has elucidated the mechanism of activity-dependent transport of ESCRT-0+ vesicles, in particular the role of kinesin-3 family member KIF13A in this process."
 - This language is too strong for the data presented, as numerous questions regarding if/how KIF13A transports ESCRT-0 vesicles remains. Is KIF13A present in axons? Does it interact with Hrs/STAM1 in neurons? Do Hrs/STAM1 directly bind KIF13A? What changes upon neuronal stimulation to either promote this interaction and/or induce KIF13A motility of Hrs/STAM1? The authors' indirect KIF13A knockdown data support this claim which is encouraging, but they are far from elucidating the mechanism. Further experiments to answer the above questions are needed for the authors to make this claim.

7. Figure 2A, 3A, 5E, and 7E kymographs do not show any motility and thus do not represent the baseline percent motility scored in the corresponding graphs. Using a more representative kymograph will help readers compare baseline to activity kymograph movement.

The following specific minor issues should be addressed:

1. Page 5: "Under control conditions, we observed that >60% of Hrs puncta were stationary over the course of the ~4-minute imaging session (Fig. 2A, D, F)."
 - In the figure and for the rest of the conditions in the text the authors provide percent motile. For consistency and ease of comparison, change the control conditions to percent motile (i.e. 40% Hrs puncta were motile).

2. Page 8: "We found that Rab35 exhibited higher motility than Hrs at baseline (~60% motile, Fig. S3D,F), but no change in motility in response to bic/4AP, even after 2h of treatment (Fig. S3D,F). However, the colocalization between RFP-Hrs and

EGFP-Rab35 in axons increased slightly but significantly in response to 2h bic/4AP (Fig. S3G-H). These data, along with similarities in the directional transport profiles of Hrs and Rab35 following 2h bic/4AP treatment (Fig. 2F and Fig. S3F), suggest that Rab35 may colocalize more strongly with the motile pool of Hrs, but that Rab35 motility is not regulated by neuronal firing in the same way."

- Is Rab35 already more associated with SV pools? Looking at Rab35 co-localization with Synapsin could be information, as it would suggest that Hrs+ vesicles (which do undergo activity-dependent trafficking) might be moving to Rab35, thus leading to the increased Hrs/Rab35 co-localization where it is needed.

3. Page 9: "However, 2h treatment with SAR405 did not alter the number or motility of axonal Hrs puncta (Fig. 5E-I), suggesting that Hrs association with vesicles occurs in the somatodendritic compartment prior to axonal transport, and remains stable in axons over the course of several hours."

- This could be tested with a longer SAR405 treatment (if neuron viability allows) or with further tracking of the Hrs-FYVE mutant to help distinguish whether the FYVE domain is important for Hrs membrane targeting to axonal vesicles or whether a different domain (such as the C-terminal coiled-coil domain brought up in the discussion) is responsible for the observed axonal Hrs puncta.

4. Page 10: "In this study, we demonstrate a novel mechanism for spatiotemporal regulation of the ESCRT pathway in neurons. Our findings show that axonal transport vesicles carrying ESCRT-0 proteins exhibit increased anterograde and bidirectional motility in response to neuronal firing (Fig. 8), providing the first example of activity-dependent transport of degradative machinery in axons."

- Is there evidence that the increased activity-dependent trafficking of Hrs/STAM1 is limited to axons? Did the authors interrogate Hrs/STAM1 dendritic trafficking?

5. Page 18: "Directionality of each puncta was categorized as follows: {greater than or equal to}4 μ m away from the cell body = anterograde, {greater than or equal to}4 μ m towards the cell body = retrograde, {greater than or equal to}4 μ m total, but <4 μ m in one direction = bidirectional, <4 μ m = stationary.

- This criteria is not clear. For each category, does "{greater than or equal to}4 μ m away" mean a net displacement of 4 μ m over the entire imaging period or during a single processive motile event? This could be further clarified by adding a supplemental figure outlining an anterograde, retrograde, bidirectional, and stationary event in a representative kymograph.

6. Figure 2: It would be helpful to label anterograde and retrograde direction above the 2A kymograph

7. Figure 6B: What does x-axis label "Distance" mean? Is it distance across the cell? Distance to the MTOC? Please define in methods or figure legend.

8. Review text for consistency with labeling of STAM1 vs STAM (or is this meaningful?) and Bicuculline/4AP vs Bic/4AP vs bic/4AP in text and figures

Reviewer #3 (Comments to the Authors (Required)):

This manuscript carries out a sophisticated cell biological analysis of axonal transport of ESCRT-0 proteins. ESCRT-0 (Hrs and STAM1 being looked at in this paper) are the first component of ESCRT complex being recruited to early endosomes to mediate generation of intraluminal vesicles in the maturing degradative pathway. This group previously showed that Hrs was required for the activity-dependent degradation of synaptic vesicle proteins which is important to remove dysfunctional SV proteins as they accumulate during activity. This is novel and we know nothing about how ESCRT-0 components are trafficked in axons to reach presynaptic sites. The authors make extensive use of live imaging in cultured rat hippocampal neurons grown in special microfluidic chambers so that axons can be separated. The authors show that Hrs-RFP is mostly stationary in axons, but increases its motility upon chemical activation (bicuculline/4AP). This increased motility occurs preferentially in the anterograde direction, giving rise to the hypothesis that activity promotes Hrs transport to presynaptic sites for improved SV degradation. There are a lot of great strengths in this paper, including live imaging, EM, CLEM, RNAi interference, pharmacological approaches. Most importantly, the authors discover one motor, KIF13A, which is involved in the SV degradative pathway. This is a significant and interesting finding. There are a number of issues that need clarification before publication, most importantly which compartment in the axon is Hrs-positive and activity-sensitive.

1) Figure 1 and 2: What is the firing behavior after acute treatment with bic/4AP vs treatment for 2 hours? Some of the experiments were done with even longer activity stimulation (48 hours). Can you explain how much synaptic activity is changed at these different treatment times? Figure 1 uses 48 hour treatment.

2) Figure 3 and 4: Are STAM1 and Hrs not transported together? The distinct motility changes after activity seem to indicate that STAM1 is different from Hrs?? Are they not on same vesicle? Have you done any dual live imaging of STAM1 and Hrs? What is the interpretation of changes in motility after acute treatment or only after 2 hours? Different mechanism implied? Different from Rab5? I think the conclusion is that Hrs+ is a subset of Rab5. I find the graphs in Figure 4 hard to parse. Is there a straight overlap count somewhere? Can you show B-G as Hrs alone, Rab5 alone, and dually positive, or is that what you are

showing? I find it very confusing to try to figure out which compartment is changing behavior. What is the Hrs+/Rab5-compartment? Is it significant? Could that be the activity-sensitive compartment? Where does Rab35 come in? Are Rab5 and Rab35 together or are they distinct vesicles?

3) Figure 5: What axonal compartment does the 2xFYVE domain associate with? What about behavior of R183A mutant in axons? I am not sure I have a clear understanding of which compartment in axons is Hrs positive and activity-sensitive.

4) Figure 6: A diagram of this assay used in Fig. 6 would be helpful to a more general readership. Since Rab5 compartments can be relocated with both KIF13A and KIF13B, but Hrs only with KIF13A, does this not imply that the relevant Hrs compartments are Rab5 negative? I don't understand what compartment you are looking at exactly.

5) Figure 7: It would be good to see some specificity/negative controls. Is KIF13B knockdown not effective for decreasing degradation? Can you rescue? Have you done more than one sh sequence? I am unclear whether or not the activity-dependent compartment is different from the baseline compartment.

F/G: Hrs vs STAM response to sh13A. Is the shCtrl + Control significantly different from the sh13A + Control? It looks like that to me. In F: only the light green bar is higher. In G: only the blue bar on the left is low, the other 3 bars look higher. This again goes back to my earlier question of Hrs and STAM are different.

Minor issues:

1) Please label all figures with the actual construct used, not just the protein name, i.e. RFP-Hrs rather than Hrs. Since almost none of the staining done is against the endogenous proteins, panels need to be labeled accordingly. It would be nice to see some endogenous staining of ESCRT in the axon, if possible. Please comment why no endogenous staining is shown.

2) In the legends, the description says ">3 weeks of replicates". What does that mean?

Response to Reviewers

We thank the reviewers for their detailed and thoughtful comments, which we feel have helped us to create a much-improved version of the manuscript. Our responses to their comments and suggestions are below in red.

Reviewer #1:

1) One widespread problem in the paper is the quality of the imaging data. The main concern of this reviewer is that many of the puncta that are depicted in kymographs cannot be followed/tracked (e.g. Fig 2B+C, 2H (what is considered a Synapsin punctum is not obvious), Fig 3A and 5F). On top of this problem, the authors routinely find 40% motility in control settings where there are basically no moving puncta shown in the representative image (which begs the question, what movement is being tracked?). For examples see Fig2A, Fig 3A, Fig5E-i and Fig 7D+E. Lastly, the quality of the imaging data is also a problem in the colocalization analyses as the images shown appear saturated (Figure 6A and C).

We have updated nearly all of the imaging data to provide better quality images. In Figure 2, we have replaced panels A-C with brighter images/kymographs of Hrs, enabling visualization of motile puncta. We have replaced the Synapsin data in former Fig. 2H with new data showing localization of Hrs with the ESCRT proteins STAM1 and CHMP4B (Fig. 2I,J). We have also replaced STAM1 images/kymographs in former Figure 3A (now Fig. S4A), Hrs images/kymographs in Figure 5E-F (now fixed immunofluorescence images in Fig. S5A), and Hrs images/kymographs in Fig. 7D-E (now in Fig. 6A,B and Fig. S8G,H). Images of transfected N2a cells in Figure 6 have also been replaced with less saturated images (now in Fig. 4B,D and Fig. S7A,C).

2) In many places in the manuscript, statistics are handled inappropriately when data are reported as "n.s." or the claim is made that a parameter didn't change. In these cases, the actual p values always need to be given: after all, a change with a $p=0.051$ is still 20 times more likely to be an actual difference than an artifact. And, even when the p value is much higher, it does not demonstrate the absence of a difference, only that the difference could not be detected. One such place, early in the text, is in Figure 2F where retrograde transport is described as not increasing with activity. The % moving retrograde clearly has increased however, but we are not informed of the magnitude, nor the p value. This is an important point since the authors make much of the fact that anterograde, but not retrograde transport is enhanced by activity. A related concern is the distinction between statistical significance and biological significance. It is true that anterograde movement is increased, but this increase is very small in actual terms of the % of Hrs puncta, especially when compared to those puncta in non-processive bidirectional movement which accounts for almost all the decrease in the stationary pool. In figure 2G, it is again unclear how likely it is that there was an acute change in colocalization since there is no p value given.

Actual p values have been added to the figure legends for all significant and non-significant comparisons, as requested.

3) The health of the neuronal cultures is assessed with vital dyes, but that is not a very sensitive method for judging neuronal health. Older microfluidic cultures can be on the decline and the authors should include some standard dissociated hippocampal cultures to determine if the motility parameters are comparable.

We have added images, kymographs, and quantification of EGFP-Hrs motility in dissociated hippocampal cultures in Fig. S3. The motility parameters are similar to those from microfluidic chambers.

4) Additional points:

1) Introduction

The premise of the work is that SV proteins need to be retrogradely transported to the soma for degradation and that ESCRT proteins need to be anterogradely transported in axons to synapses so that the initial steps for SV protein degradation can occur locally. The introduction asserts these premises largely without citations and indeed some of these premises are on shaky ground. There are lysosomes in axons, there is retrograde transport of SVs which could permit quality control in the soma without ESCRT in the periphery, and there is ESCRT present already in the periphery so that additional activity-mediated transport may not be essential. Truckenbrodt, for example, found that some older synaptotagmin was found in the soma, but also that inhibition of lysosomes caused a local increase in synaptotagmin, implying local degradation as well as retrograde transport.

We have added references to support these concepts, and we certainly agree that ESCRT-mediated retrograde transport is not the only degradative mechanism for SV proteins. While it is the case that lysosomes and lysosome-like structures can be detected in axons, multiple studies have shown that degradative lysosomes are primarily located in the cell body (see Cai et al., 2010; Gowrishankar et al. 2015 and 2017; Yap et al. 2018). Moreover, disruption of lysosomal proteolysis inhibits the retrograde transport of late endosomes (Lee et al., *J Neurosci*, 2011), so the data from Truckenbrodt et al. (*EMBO J*, 2018) are consistent with the idea of retrograde transport to the soma being required for synaptotagmin degradation.

2) Figure 1.

--Concerning the representative pictures in 1a and 1b, it would be nice if they showed representative images for the TTX and gabazine treatments (at least for the endosomes). It is my understanding that the pictures in 1A and 1B are just to show what they are scoring as endosomes and multilamellar structures.

We now include representative images for TTX and gabazine treatments (Fig. 1A,B).

--it would be preferable to show actual p-values rather than stars or ns (Fig 1c and 1D) even if the $P < 0.0001$ as written in the figure legend.

The actual p values have been added.

3) Figure 2.

--It is hard to see Hrs puncta in 2C and the still image does not quite match up with the kymograph. Also, please indicate the time span of the kymograph as well the acquisition rate (fps) of the time lapse (for all time lapse experiments).

We have replaced this image/kymograph, and we now indicate time spans and acquisition rates for all time lapse experiments in the figure legends.

--In figure 2h instead of performing colocalization of 2 overexpressed proteins, it would be preferable to determine the colocalization of the endogenous proteins. More specifics on how % time colocalized was determined would also be appreciated.

We have replaced the data in Figure 2H, which we agree was difficult to interpret, with panels showing Hrs colocalization with other ESCRT proteins. We now examine the % colocalization of Hrs with Synapsin in fixed axons following 20-hour DMSO or Bic/4AP treatment (Fig. 7), to demonstrate that KIF13A is required for the activity-dependent transport of Hrs to synaptic vesicle pools. We co-transfected the Hrs and Synapsin constructs in order to more definitively verify colocalization (with the endogenous proteins, it is often difficult to assign puncta to specific axons).

4) Figure S1.

- S1 A. Do the neurons seeded in the tripartite chamber efficiently form synapses at DIV13-15?

Yes, synapse formation occurs in these chambers, and we now include references for these studies (p. 6). We have also verified synapse formation in the chambers via immunostaining, but we do not include these data here due to space constraints and the previously published characterization of these chambers.

--S1 B, C. Please clarify how total number of neurons were counted in order to calculate % transduced cells (e.g. DIC, Hoechst). If dye was used, were glia eliminated from count?

We used DAPI to count the total number of cells, and have clarified this in the figure legend. Glia were not eliminated from this count, but our embryonic hippocampal cultures typically have <10% glial cells.

-- S1 D. the data spread for the control is somewhat concerning because it looks bimodal and the n is pretty low. It is also concerning that it there appears there is no difference in motility between the TTX and Bic/4AP samples.

We have increased the 'n' for these experiments. There is a significant difference in motility between the TTX and Bic/4AP conditions (new Fig. S2D).

5) Figure 4.

--The authors should show a better representative picture for Hrs (Fig 4A).

We now show better representative images for Hrs and Rab5 (new Figure 3G).

6) Figure 7. I understand that SV2 is blurry on blots, but the Western blot does not instill confidence in the argument the authors are trying to make.

We have rerun these blots, which are shown in the new Figure 7E. We have also used another method, SNAPtag pulse-chase (Fig. 7H,I), to verify the impairment in VAMP2 protein degradation seen in the cycloheximide-chase assay.

7) Minor point. The way the IP is described for Fig 6 needs to be toned down: "Consistent with our transport assays, we found that Hrs clearly precipitated KIF13A, but not KIF13B (Fig. 6E-F). STAM1 also had some ability to precipitate KIF13A, albeit variably, but did not precipitate KIF13B (Fig. 6E-F)." However, in 6E, there is a clear Myc band in the IP for both Stam and Hrs (albeit less than Kif13A but it is still very much visible to the reader).

We agree with these comments. To reflect the fact that there is some interaction between Hrs and KIF13B, we have now quantified the amount of KIF13A and KIF13B precipitated by Hrs (Figure 4H), and included quantification for the linker-induced transport of Hrs by KIF13A and KIF13B (Figure 4F). At the same time, we have decided to focus our revised study on Hrs in the interest of space, and have

thus removed the STAM1 IP data.

Reviewer #2:

1. Page 5: "Under control conditions, we observed that >60% of Hrs puncta were stationary over the course of the ~4-minute imaging session (Fig. 2A, D, F). Acute treatment with bic/4AP led to a slight but significant increase in puncta motility (from ~38% to ~50% motile), with the effect growing stronger after 2h of treatment (~60% motile puncta; Fig. 2A-D, F)."

- The observed increase in bidirectional motility doesn't inform the reader if the movement is ultimately processive. Net displacement measurements are needed to demonstrate processive anterograde trafficking.

We have now analyzed net displacement, and find that the total net displacement of Hrs puncta shifts in the anterograde direction after 2-hour Bic/4AP treatment (Figure 1G,H).

Additionally, the kymographs of Hrs motility in Figure 2B, C, and H do not support the idea that Hrs undergoes deliberate anterograde trafficking upon neuron firing. Additional analysis of the movement and whether it ultimately leads to co-localization with SV pools (as suggested below in point 2) is needed to support the authors conclusion that the Hrs trafficking is important for SV degradation.

Per the reviewer's suggestion, we now look at Hrs trafficking to SV pools over a longer time period, based on its colocalization with Synapsin after 20-hour treatment with DMSO or Bic/4AP (Figure 7A-D). We find that 1) Hrs colocalization with Synapsin increases following Bic/4AP treatment, indicative of increased activity-dependent transport of Hrs to SV pools (Fig. 7A-D), and 2) knockdown of KIF13A prevents this phenomenon (Fig. 7A-D) and slows SV protein degradation (Fig. 7E-I), demonstrating that KIF13A-mediated transport is required for Hrs delivery to SV pools and subsequent SV protein degradation.

2. Page 6: "While we did observe motile Hrs puncta colocalizing with Synapsin at presynaptic sites (Fig. 2G-H, Video 3), this behavior did not depend upon neuronal activity, as motile Hrs puncta spent ~35% of their time in contact with Synapsin regardless of activity condition."

- This is concerning, as the proposed role of the trafficking is to promote activity-dependent degradation of SV pools. Are there more representative kymographs that display Hrs moving to a site of Synapsin and pausing? The authors' live imaging data could be mined to determine the percent of Hrs/STAM1 puncta that move to and pause at Synapsin sites. This type of analysis of ESCRT-0 protein movement could be very informative and support their conclusions.

As discussed above, we have now shifted our analysis to measuring Hrs/Synapsin colocalization following a 20-hour time course of Bic/4AP treatment (Fig. 7). Our analyses of the data in Fig. 2G-H suggested to us that acute or 2-hour Bic/4AP treatment did not capture the relevant time frame for monitoring Hrs trafficking to presynaptic sites. Moreover, we found that Hrs/Synapsin interactions were difficult to characterize in our 4-min imaging videos, as Hrs pausing at Synapsin puncta was often followed by movement away from these sites, suggesting that Hrs+ vesicles had not yet arrived at their final destinations.

3. Page 6: "However, Hrs/Synapsin colocalization was significantly increased following 2h of treatment with PR619, a deubiquitinating enzyme inhibitor that promotes SV protein ubiquitination (Fig. S1E-F), suggesting that Hrs interaction with presynaptic sites is ubiquitin-dependent."

- This is very interesting, but as the increased co-localization was not observed with the bic-4AP treatment it opens up additional questions. Looking at VAMP2-ubiquitination (as in Supplemental Figure

1E) under various neuronal stimulation conditions could help link this finding to the idea that neuronal activity induces SV protein ubiquitination and subsequent ESCRT-dependent degradation.

This is an excellent suggestion, and we did in fact observe increased VAMP2 ubiquitination in neurons following 2h Bic/4AP treatment (see below). However, we elected to remove these experiments from the manuscript with the recognition that DUB inhibition likely causes many effects in neurons and at the synapse, and we would therefore need to use additional methods to directly link SV protein ubiquitination to Hrs pausing/ interaction with SV pools. Instead, we have focused the manuscript on the mechanism of activity-dependent trafficking of Hrs, and provided more data to support a role for KIF13A in this process.

4. Page 6: "Intriguingly, Hrs-positive structures identified in the axon appear to be degradative in nature, including an endosome and a multivesicular body (Fig. S2). These data, along with our previous findings (Sheehan et al. 2016), suggest that Hrs undergoes activity-dependent transport in axons and is associated with degradative structures.

Page 11: "Moreover, our CLEM experiments demonstrate that Hrs+ vesicles are degradative structures (Fig. S2), though we cannot yet correlate their morphology with their axonal transport history to determine what stage of the degradative process they represent."

- Based solely on the EM data, the authors cannot state that the Hrs+ structures are degradative. Follow-up experiments are needed to confirm this statement.

We agree with the reviewer, and have changed this language to: "we found Hrs labeling associated with endosomal structures including an MVB (Fig. 1G, arrows), confirming its presence at putative sites of endolysosomal sorting."

5. Page 8: " Interestingly, while the 2xFYVE domain exhibited highly punctate expression in axons, its dynamics were different from those of full-length Hrs and it did not demonstrate activity-dependent changes in motility (Fig. S4A-F), indicating that other domains of Hrs facilitate its activity-dependent transport.

- This study would greatly benefit from additional Hrs domain analysis. If the authors could identify the domain that is responsible for the activity-dependent trafficking, and better yet link the domain to an association, either direct or indirect, with KIF13A, the conclusions of this paper would be more strongly supported.

We agree that such domain analysis would be informative, but were told by the editor that it would not be necessary to pursue this strategy for the paper revision.

6. Page 12: "Finally, our work has elucidated the mechanism of activity-dependent transport of ESCRT-0+ vesicles, in particular the role of kinesin-3 family member KIF13A in this process."

- This language is too strong for the data presented, as numerous questions regarding if/how KIF13A transports ESCRT-0 vesicles remains. Is KIF13A present in axons? Does it interact with Hrs/STAM1 in neurons? Do Hrs/STAM1 directly bind KIF13A? What changes upon neuronal stimulation to either promote this interaction and/or induce KIF13A motility of Hrs/STAM1? The authors' indirect KIF13A knockdown data support this claim which is encouraging, but they are far from elucidating the mechanism. Further experiments to answer the above questions are needed for the authors to make this claim.

We now include additional experiments that provide evidence for KIF13A in axons (immunostaining in Figure S8C) and for an activity-dependent interaction between Hrs and KIF13A (new Figure 5). For the latter experiments, we used the proximity ligation assay (PLA) to look at interactions between EGFP-Hrs and endogenous KIF13A in neuronal cell bodies under control and 2-hour Bic/4AP conditions. We found that PLA puncta increased significantly following Bic/4AP treatment, indicative of an activity-dependent interaction between Hrs and KIF13A (Fig. 5A-C). We also performed coimmunoprecipitation of mCh-Hrs and KIF13A-FRB-Myc in N2a cells treated with control or high K⁺ Tyrodes solution to mimic neuronal activity. Here, we found that Hrs precipitated significantly more KIF13A in the high K⁺ condition (Fig. 5D,E), again suggesting that the Hrs/KIF13A interaction is activity-dependent.

7. Figure 2A, 3A, 5E, and 7E kymographs do not show any motility and thus do not represent the baseline percent motility scored in the corresponding graphs. Using a more representative kymograph will help readers compare baseline to activity kymograph movement.

We have replaced these images and kymographs with better representative images -- see our response to the same issue raised by Reviewer 1 (p. 1).

The following specific minor issues should be addressed:

1. Page 5: "Under control conditions, we observed that >60% of Hrs puncta were stationary over the course of the ~4-minute imaging session (Fig. 2A, D, F)."

- In the figure and for the rest of the conditions in the text the authors provide percent motile. For consistency and ease of comparison, change the control conditions to percent motile (i.e. 40% Hrs puncta were motile).

We have made this change to the text.

2. Page 8: "We found that Rab35 exhibited higher motility than Hrs at baseline (~60% motile, Fig. S3D,F), but no change in motility in response to bic/4AP, even after 2h of treatment (Fig. S3D,F). However, the colocalization between RFP-Hrs and EGFP-Rab35 in axons increased slightly but significantly in response to 2h bic/4AP (Fig. S3G-H). These data, along with similarities in the directional transport profiles of Hrs and Rab35 following 2h bic/4AP treatment (Fig. 2F and Fig. S3F), suggest that Rab35 may colocalize more strongly with the motile pool of Hrs, but that Rab35 motility is not regulated by neuronal firing in the same way."

- Is Rab35 already more associated with SV pools? Looking at Rab35 co-localization with Synapsin could be information, as it would suggest that Hrs+ vesicles (which do undergo activity-dependent trafficking) might be moving to Rab35, thus leading to the increased Hrs/Rab35 co-localization where it is needed.

We have followed the reviewer's insightful suggestion and looked at Rab35 colocalization with Synapsin (Fig. S6F,G). Indeed, 80% of mCh-Rab35 puncta colocalize with EGFP-Synapsin, and this value does not change following Bic/4AP treatment, supporting the reviewer's theory that Rab35 is already associated with SV pools. However, in our efforts to obtain better images for our Rab35/Hrs colocalization experiments (old Fig. S3G), we noticed that Rab35 overexpression seemed to alter Hrs transport dynamics. Our subsequent attempt to measure Hrs/Rab35 colocalization by immunostaining was also ambiguous due to the high background of the Rab35 antibody in axons (see below). Therefore, we have removed the Hrs/Rab35 colocalization data from the manuscript, and instead we now show the characterization of Rab35 motility (Fig. S6A-E) along with mCh-Rab35 colocalization with EGFP-Synapsin (Fig. S6F,G). These data demonstrate that Rab35 axonal transport is unaffected by neuronal activity, in contrast to Hrs transport.

3. Page 9: "However, 2h treatment with SAR405 did not alter the number or motility of axonal Hrs puncta (Fig. 5E-I), suggesting that Hrs association with vesicles occurs in the somatodendritic compartment prior to axonal transport, and remains stable in axons over the course of several hours."

- This could be tested with a longer SAR405 treatment (if neuron viability allows) or with further tracking of the Hrs-FYVE mutant to help distinguish whether the FYVE domain is important for Hrs membrane targeting to axonal vesicles or whether a different domain (such as the C-terminal coiled-coil domain brought up in the discussion) is responsible for the observed axonal Hrs puncta.

We have now performed experiments to examine the effects of 24-hour SAR405 treatment on the density and intensity of Hrs puncta in axons. Interestingly, this longer treatment disrupted EGFP-Hrs puncta density and intensity to a similar extent as the R183A FYVE domain mutant (new Fig. S5A-C). These data suggest that the FYVE domain/PI3P interaction is indeed important for Hrs association with axonal vesicles.

4. Page 10: "In this study, we demonstrate a novel mechanism for spatiotemporal regulation of the ESCRT pathway in neurons. Our findings show that axonal transport vesicles carrying ESCRT-0 proteins exhibit increased anterograde and bidirectional motility in response to neuronal firing (Fig. 8), providing the first example of activity-dependent transport of degradative machinery in axons."

- Is there evidence that the increased activity-dependent trafficking of Hrs/STAM1 is limited to axons? Did the authors interrogate Hrs/STAM1 dendritic trafficking?

We are currently investigating this question, but it is beyond the scope of the current study.

5. Page 18: "Directionality of each puncta was categorized as follows: {greater than or equal to}4 μm away from the cell body = anterograde, {greater than or equal to}4 μm towards the cell body = retrograde, {greater than or equal to}4 μm total, but <4 μm in one direction = bidirectional, <4 μm = stationary.

- This criteria is not clear. For each category, does "{greater than or equal to}4 μm away" mean a net displacement of 4 μm over the entire imaging period or during a single processive motile event? This could be further clarified by adding a supplemental figure outlining an anterograde, retrograde, bidirectional, and stationary event in a representative kymograph.

The ' $\geq 4 \mu\text{m}$ ' refers to net displacement over the course of the imaging period. We have clarified this point in the Methods section (p. 22) and added a kymograph showing examples of anterograde, retrograde, and stationary events (Fig. S2E).

6. Figure 2: It would be helpful to label anterograde and retrograde direction above the 2A kymograph.

We have added these labels.

7. Figure 6B: What does x-axis label "Distance" mean? Is it distance across the cell? Distance to the MTOC? Please define in methods or figure legend.

Distance = number of pixels from the origin of the yellow line (moving left to right) in merged images (Fig. 4B,D). This has been clarified in the figure legend.

8. Review text for consistency with labeling of STAM1 vs STAM (or is this meaningful?) and Bicuculline/4AP vs Bic/4AP vs bic/4AP in text and figures.

We have standardized our labeling for STAM1 and Bic/4AP in the text and figures.

Reviewer #3:

1) Figure 1 and 2: What is the firing behavior after acute treatment with bic/4AP vs treatment for 2 hours? Some of the experiments were done with even longer activity stimulation (48 hours). Can you explain how much synaptic activity is changed at these different treatment times? Figure 1 uses 48 hour treatment.

Several different treatments were used to induce neuronal firing in our study. In Figure 1, hippocampal neurons were treated with the GABA receptor antagonist gabazine, which, similar to bicuculline, significantly increases neuronal firing rates upon application. This increased firing triggers homeostatic plasticity/synaptic scaling mechanisms that gradually return firing rates to baseline over the course of 48 hours (Turrigiano et al., 1998). Thus, the biggest changes in neuronal activity are induced by gabazine within the first few hours of treatment. However, we hypothesized that it would take time for neuronal activity to induce Hrs transport to presynaptic terminals, leading to the recruitment of downstream ESCRT proteins and MVB formation, etc., and therefore we decided to look at endosome/MVB formation in these neurons after 48 hours of gabazine treatment (around the time that firing should be returning to baseline). In our subsequent experiments, we used bicuculline together with the voltage-activated K⁺ channel blocker 4-aminopyridine (Bic/4AP) to increase neuronal firing over shorter (~5 min to 20 hour) time periods. The efficacy of this treatment has been previously documented by us (Sheehan et al., *J Neurosci*, 2016) and others (e.g. Li, Popko, et al. *J Neurophysiol*, 2015). We performed imaging experiments to measure Hrs motility/axonal transport shortly after drug application (5 min – 2hr), when neuronal firing is high. When measuring Hrs trafficking to SV pools, we looked after 20h of Bic/4AP treatment. Although neuronal firing rates are presumably not as high at this later timepoint, we chose it to allow time for trafficking of Hrs transport vesicles to their ultimate destinations.

We acknowledge that application of Bic/4AP promotes intense firing that does not represent a physiological condition. We chose this treatment for consistency, given that we had previously used it for our experiments in Sheehan et al. We have now also measured Hrs motility in the presence of brain-derived neurotrophic factor (BDNF), shown to rapidly enhance excitatory neurotransmission in cultured hippocampal neurons (Lessman et al., *Neuroreport*, 1994; Lessman, *Gen Pharmacol*, 1998; Levine et al., *PNAS*, 1995; Levine et al., *J Neurosci*, 1996) and used to stimulate neuronal activity in our recent study with Dr. Francesca Bartolini's lab (Qu et al., *Curr Biol*, 2019). Indeed, we find that Hrs motility is similarly increased by 1 hour treatment with BDNF (new Fig. S3H-J), and that this effect is prevented by knockdown of KIF13A (new Fig. S8H,I), indicating that Hrs transport is elicited by other stimuli that induce neuronal firing, in a KIF13A-dependent manner.

2) Figure 3 and 4: Are STAM1 and Hrs not transported together? The distinct motility changes after activity seem to indicate that STAM1 is different from Hrs?? Are they not on same vesicle? Have you done any dual live imaging of STAM1 and Hrs?

We have now performed live imaging of co-expressed Hrs and STAM (Fig. 1I). Indeed, they are transported primarily on the same vesicles (80% colocalization), in contrast to Hrs and the ESCRT-III protein CHMP4B, which are transported on largely distinct vesicles (~20% colocalization)(Fig. 1J, K). However, we do observe some differences in motility for singly-expressed STAM1 vs. Hrs (*e.g.* STAM1 exhibits increased retrograde transport in response to Bic/4AP treatment; see Fig. S4F). Whether these differences reflect the true biology of these proteins, or the fact that STAM1 is more efficiently/highly expressed in neurons than Hrs due to its smaller size, is unclear.

What is the interpretation of changes in motility after acute treatment or only after 2 hours? Different mechanism implied?

Our interpretation is that changes in Hrs motility after acute and 2-hour Bic/4AP treatment are due to the same mechanism, but that the magnitude of the effect is larger after 2 hours.

Different from Rab5? I think the conclusion is that Hrs+ is a subset of Rab5. I find the graphs in Figure 4 hard to parse. Is there a straight overlap count somewhere? Can you show B-G as Hrs alone, Rab5 alone, and dually positive, or is that what you are showing?

Indeed, our conclusion is that Hrs+ vesicles represent a subset of Rab5+ vesicles. We have simplified our presentation of these data (new Figure 3) to characterize: 1) the % of motile Hrs+ vesicles that are also Rab5+ (panel H), 2) the % of motile Rab5+ vesicles that are also Hrs+ (panel I), and 3) the density of motile Hrs+/Rab5+ puncta per length axon (panel J).

I find it very confusing to try to figure out which compartment is changing behavior. What is the Hrs+/Rab5- compartment? Is it significant? Could that be the activity-sensitive compartment?

We do not specifically examine Hrs+/Rab5- vesicles. We show that ~80% of motile Hrs+ vesicles are also Rab5+, and that the total number of Hrs+/Rab5+ vesicles increases significantly following Bic/4AP treatment (Fig. 3G-J), indicating that the Hrs+ subset of Rab5 vesicles is responsive to activity.

Where does Rab35 come in? Are Rab5 and Rab35 together or are they distinct vesicles?

We did not perform co-imaging of Rab5 and Rab35, but based on their differences in activity-induced motility (Fig. 3 for Rab5; Fig. S6 for Rab35), we suspect that they are on distinct vesicles.

3) Figure 5: What axonal compartment does the 2xFYVE domain associate with? What about behavior of R183A mutant in axons? I am not sure I have a clear understanding of which compartment in axons is Hrs positive and activity-sensitive.

Neither the 2xFYVE domain nor the Hrs R183A mutant recapitulate the properties of full-length, wild-type Hrs. Specifically, the 2xFYVE domain does not exhibit activity-induced motility (Fig. S5D-G), and the R183A mutant has a more diffuse distribution than the wild-type protein (Fig. S5A-C). These findings demonstrate that the FYVE domain is important for Hrs association with vesicles, but is not sufficient to confer activity-dependent motility – presumably because the Hrs/KIF13A interaction is mediated by a different domain of Hrs.

4) Figure 6: A diagram of this assay used in Fig. 6 would be helpful to a more general readership. Since Rab5 compartments can be relocated with both KIF13A and KIF13B, but Hrs only with KIF13A, does this not imply that the relevant Hrs compartments are Rab5 negative? I don't understand what compartment you are looking at exactly.

We have added a diagram to this figure (new Figure 4A) as requested. Our results with the KIF13A/B cargo transport assay suggest that Hrs would colocalize with Rab5+ vesicles that are transported by KIF13A, but not those transported by KIF13B. However, given that this assay is designed solely to investigate transport and requires overexpression of the cargo binding domain of KIF13A/B as well as the dynein binding domain of BicD2, we would not feel comfortable using it to characterize the nature of Hrs+ or Rab5+ compartments.

5) Figure 7: It would be good to see some specificity/negative controls. Is KIF13B knockdown not effective for decreasing degradation? Can you rescue? Have you done more than one sh sequence? I am unclear whether or not the activity-dependent compartment is different from the baseline compartment.

These are excellent suggestions, and we have now repeated the Hrs imaging experiments with a second shRNA against KIF13A (shKIF13A2) and an shRNA against KIF13B (shKIF13B). We find that KIF13B knockdown does not impact activity-induced axonal transport of Hrs, while both shRNAs against KIF13A prevent this transport (Fig. 6, Fig. S8). Moreover, both KIF13A shRNAs block the activity-dependent delivery of Hrs to SV pools (Fig. 7A-D) and slow the degradation of VAMP2 (Fig. 7H,I).

F/G: Hrs vs STAM response to sh13A. Is the shCtrl + Control significantly different from the sh13A + Control? It looks like that to me. In F: only the light green bar is higher. In G: only the blue bar on the left is low, the other 3 bars look higher. This again goes back to my earlier question of Hrs and STAM are different.

As discussed above, it is possible that Hrs and STAM have slightly different patterns of motility. With regard to our previous Figure 7F/G, we observed a different baseline motility of STAM1 under shCtrl vs. shKIF13A conditions. This difference may reflect KIF13A regulation of STAM1 transport under baseline conditions, or differences in the cultures used for shCtrl and shKIF13A experiments (since we only compared +/- Bic/4AP within each shRNA condition, the experiments were performed in different batches of neurons). In the revised manuscript, we decided to focus our efforts on characterizing Hrs transport, so we have removed these STAM1 data due to space limitations.

Minor issues:

1) Please label all figures with the actual construct used, not just the protein name, i.e. RFP-Hrs rather than Hrs. Since almost none of the staining done is against the endogenous proteins, panels need to be labeled accordingly. It would be nice to see some endogenous staining of ESCRT in the axon, if possible. Please comment why no endogenous staining is shown.

We have made the requested change to names of constructs used. We now include immunostaining against endogenous Hrs, and have verified that the endogenous protein is recruited into axons by 20-hour Bic/4AP treatment (Fig. S1).

2) In the legends, the description says ">3 weeks of replicates". What does that mean?

Data comes from >3 different batches (cultures) of neurons prepared in different weeks. This has been clarified in the figure legends.

May 16, 2022

RE: Life Science Alliance Manuscript #LSA-2020-00745-TR

Dr. Clarissa Waites
Columbia University Medical Center
630 W. 168th St.
New York, NY 10032

Dear Dr. Waites,

Thank you for submitting your revised manuscript entitled "Transport of Hrs is activity-dependent and rate limiting for synaptic vesicle protein degradation". We would be happy to publish your paper in Life Science Alliance pending final revisions necessary to meet our formatting guidelines.

- please address the remaining Reviewer 1's concerns by removing claims about directional transport and add more suitable caveats and improvements in displaying movies and kymographs
- please upload your main and your supplementary figure files as single files
- please add the Twitter handle of your host institute/organization as well as your own or/and one of the authors in our system
- please use the [10 author names, et al.] format in your references (i.e. limit the author names to the first 10)
- please add the supplementary figure and video legends to the main manuscript text
- please double-check your figure-callouts; on page 13, you have a callout for Figure 8 H,I, but these do not exist in the legend or the panel; please add a callout for Figure S2E

Figure Check:

-Figure 1G: is bottom left zoomed in supposed to correlate with boxes that are on top of 1G? The images do not seem to match; for the images to the right, there should be insets to indicate which parts are zoomed in.

A. FINAL FILES:

B. MANUSCRIPT ORGANIZATION AND FORMATTING:

Sincerely,

Reviewer #1 (Comments to the Authors (Required)):

The authors have cleared up many issues and there is a lot of data of interest in the manuscript concerning Hrs and Kif13. I regret to say I do not find the author's responses and changes to be compelling on a key point - the case for a net change in HRS transport remains very weak. I feel that they continue to make a mountain out of the very small molehill that is the difference between anterograde and retrograde transport in Figure 2F-H. The images from fixed tissue are indeed improved and brighter, but the videos are still very difficult to score. To my eye there is a great range of brightness that extends from very clear puncta to puncta that are almost indistinguishable from noise. This makes any quantification of movement pretty much impossible; 90% of the movement of HRS could be invisible and excluded from the analysis. Looking at the videos I could not decide what they considered a particle and what they did not. It would certainly help to have arrows marking what they judged to be real, but if that is difficult judgment call, the concern that they are only scoring a tiny fraction of the Hrs movement would remain. In addition, the kymographs in Fig 2 and in S3 don't show any appreciable level of net motility. Almost all particles are stationary, with just a few brief, short range, movements. Where the percent motility has increased, e.g. S3H there is no information on directionality and the kymographs suggest it has no meaningful directional bias. The evidence that most Hrs is co-transported with STAM1 and that STAM1 transport is increased in both directions with the same trends as Hrs transport also undercuts the idea that there is a meaningful bias towards delivery of new Hrs to synapses and that this is rate determining. I think the manuscript would be stronger if the claims about directional transport were omitted and there were more suitable caveats and improvements in displaying movies and kymographs. The rest of the valuable descriptive data can speak for itself.

Reviewer #3 (Comments to the Authors (Required)):

This is a revised manuscript, strengthened by more data and clearer writing. The data are well done with extensive quantification. The main findings are important and contribute to new conceptualization of activity-dependent trafficking of ESCRT machinery for synaptic vesicle turnover in axon terminals. I recommend publication.

Response to editor and reviewer comments

We have addressed the comments as indicated below **in red**.

Editor comments:

-please address the remaining Reviewer 1's concerns by removing claims about directional transport and add more suitable caveats and improvements in displaying movies and kymographs

We have addressed these comments – please see our more detailed response to Reviewer 1's comments below.

-please upload your main and your supplementary figure files as single files

All files are prepared for uploading as specified.

-please add the Twitter handle of your host institute/organization as well as your own or/and one of the authors in our system

OK.

-please use the [10 author names, et al.] format in your references (i.e. limit the author names to the first 10)

We have changed the reference format in our manuscript as requested.

-please add the supplementary figure and video legends to the main manuscript text

Supplementary figure and video legends have been added to the manuscript as requested.

-please double-check your figure-callouts; on page 13, you have a callout for Figure 8 H,I, but these do not exist in the legend or the panel; please add a callout for Figure S2E

We have corrected these mistakes.

Figure Check:

-Figure 1G: is bottom left zoomed in supposed to correlate with boxes that are on top of 1G? The images do not seem to match; for the images to the right, there should be insets to indicate which parts are zoomed in.

We have clarified this issue in the figure legend for Figure 1.

Reviewer 1 comments

The authors have cleared up many issues and there is a lot of data of interest in the manuscript concerning Hrs and Kif13. I regret to say I do not find the author's responses and changes to be compelling on a key point - the case for a net change in HRS transport remains very weak. I feel that they continue to make a mountain out of the very small molehill that is the difference between anterograde and retrograde transport in Figure 2F-H. The images from fixed tissue are indeed improved and brighter, but the videos are still very difficult to score. To my eye there is a great range of brightness that extends from very clear puncta to puncta that are almost indistinguishable from noise. This makes any quantification of movement pretty much impossible; 90% of the movement of HRS could be invisible and excluded from the analysis. Looking at the videos I could not decide what they

considered a particle and what they did not. It would certainly help to have arrows marking what they judged to be real, but if that is difficult judgment call, the concern that they are only scoring a tiny fraction of the Hrs movement would remain. In addition, the kymographs in Fig 2 and in S3 don't show any appreciable level of net motility. Almost all particles are stationary, with just a few brief, short range, movements. Where the percent motility has increased, e.g. S3H there is no information on directionality and the kymographs suggest it has no meaningful directional bias. The evidence that most Hrs is co-transported with STAM1 and that STAM1 transport is increased in both directions with the same trends as Hrs transport also undercuts the idea that there is a meaningful bias towards delivery of new Hrs to synapses and that this is rate determining. I think the manuscript would be stronger if the claims about directional transport were omitted and there were more suitable caveats and improvements in displaying movies and kymographs. The rest of the valuable descriptive data can speak for itself.

The reviewer makes some good points, and we agree that it is difficult to make a compelling case for the long-range directional transport of Hrs. There is considerable variability in Hrs puncta size and brightness, and much of its transport occurs in bursts rather than as continuous, unidirectional movement throughout the imaging session. Rab5 puncta exhibit similar behavior, indicating that this pattern of motility is common for early endosomes. We have therefore taken the reviewer's suggestion to remove or soften our claims about directional transport of Hrs, by making the following changes:

1) In the Results section, we have edited the text to clarify the parameters used to measure anterograde, retrograde, and bidirectional movement of Hrs in Figure 2 (p. 7). We have also softened our claim about Hrs preferentially undergoing anterograde transport, and moved the net displacement analyses to Supplemental Figure 3 in order to de-emphasize the significance of this slight shift towards anterograde movement (p. 7).

2) We have changed the title to "Axonal transport of Hrs is activity-dependent and facilitates synaptic vesicle protein degradation" in order to address the reviewer's comment that Hrs transport is not rate determining.

3) In the Discussion, we have added a section discussing the possibility of Hrs undergoing activity-dependent retrograde transport, similar to STAM1, and the implications of such transport for SV protein degradation (p. 15).

May 19, 2022

RE: Life Science Alliance Manuscript #LSA-2020-00745-TRR

Dr. Clarissa Waites
Columbia University Medical Center
630 W. 168th St.
Black Building 1210B
New York, NY 10032

Dear Dr. Waites,

Thank you for submitting your Research Article entitled "Axonal transport of Hrs is activity-dependent and facilitates synaptic vesicle protein degradation". It is a pleasure to let you know that your manuscript is now accepted for publication in Life Science Alliance. Congratulations on this interesting work.

DISTRIBUTION OF MATERIALS:

Again, congratulations on a very nice paper. I hope you found the review process to be constructive and are pleased with how the manuscript was handled editorially. We look forward to future exciting submissions from your lab.

Sincerely,
